# Unlearning's Blind Spots:
# Over-Unlearning and Prototypical Relearning Attack

**SeungBum Ha** [1]  **Saerom Park** [1 2]  **Sung Whan Yoon** [1 3]

## Abstract

Machine unlearning (MU) aims to expunge a designated forget set from a trained model without costly retraining, yet the existing techniques overlook two critical blind spots: "over-unlearning" that deteriorates retained data near the forget set, and post-hoc "relearning" attacks that aim to resurrect the forgotten knowledge. Focusing on class-level unlearning, we first derive an over-unlearning metric, $OU@\varepsilon$, which quantifies collateral damage in regions proximal to the forget set, where over-unlearning mainly occurs. Next, we expose an unforeseen relearning threat on MU, i.e., the Prototypical Relearning Attack, which exploits the per-class prototype of the forget class with just a few samples, and easily restores the pre-unlearning performance. To counter both blind spots in class-level unlearning, we introduce `Spotter`, a plug-and-play objective that combines (i) a masked knowledge-distillation penalty on the nearby region of forget classes to suppress $OU@\varepsilon$, and (ii) an intra-class dispersion loss that scatters forget-class embeddings, neutralizing Prototypical Relearning Attacks. `Spotter` achieves state-of-the-art results across CIFAR, TinyImageNet, and CASIA-WebFace datasets, offering a practical remedy to unlearning's blind spots.

## 1. Introduction

With the rise of the deep learning era, explosive growth in artificial intelligence raises concerns regarding its side effects, particularly in relation to regulatory acts such as GDPR (Voigt & von dem Bussche, 2017) and CCPA (Goldman, 2020), as well as fundamental human rights like the *"Right to be Forgotten."* For instance, large-scale datasets such as LAION-5B (Schuhmann et al., 2022) have faced issues related to the presence of Child Sexual Abuse Material (CSAM) (Thiel, 2023). These issues raise not only legal and ethical concerns but also the risk of criminal liability and secondary victimization, while addressing them typically requires identifying and removing controversial data or knowledge, followed by costly model retraining. To tackle these challenges more effectively, Machine Unlearning (MU) has recently emerged as a remedy, aiming to eliminate the model's reliance on specified training data or target classes/concepts to address privacy violations and related ethical concerns (Shaik et al., 2025).

The recent achievements of MU have shown the technical validity of forgetting knowledge; however, recent studies have also highlighted a recurring failure mode–"over-unlearning"–where unlearning overshoots and inadvertently degrades retained performance and generalization (Hu et al., 2024a). Such collateral damage is often most pronounced for samples near or similar to the forget set, e.g., around entangled decision boundaries. Nevertheless, the community still lacks a principled evaluation protocol for quantifying over-unlearning in class-level unlearning, where collateral damage tends to concentrate around the forget classes.

Meanwhile, a practical risk arises after unlearning: an adversary may supply only a handful of "forgotten" examples and rapidly re-induce the removed knowledge. Such relearning attacks have been demonstrated against large language models (Lynch et al., 2024; Deeb & Roger, 2024; Hu et al., 2025), where a small set of forget prompts can revive banned content. Despite these advances, relearning threats in class-centric vision tasks remain largely underexplored. In practical deployments–such as face recognition and identity-level content moderation–adversaries can readily harvest a handful of personal images from social media or public profiles, making systematic investigation in visual classification both urgent and necessary. We show that current unlearning methods for image classification models also remain susceptible by introducing our new, highly efficient Prototypical Relearning Attack (PRA) that succeeds with

[1]Graduate School of Artificial Intelligence, Ulsan National Institute of Science and Technology (UNIST), Ulsan, South Korea [2]Department of Industrial Engineering, UNIST, Ulsan, South Korea [3]Department of Electrical Engineering, UNIST, Ulsan, South Korea. Correspondence to: Saerom Park <srompark@unist.ac.kr>, Sung Whan Yoon <shyoon8@unist.ac.kr>.

*Proceedings of the 43rd International Conference on Machine Learning*, Seoul, South Korea. PMLR 306, 2026. Copyright 2026 by the author(s).

a few examples per forget class. It computes an averaged per-class feature with a few forgotten samples and utilizes it as the corresponding classifier of the relearned model.

In this paper, we propose a practical remedy for shedding light on these two blind spots. To tackle the over-unlearning issue, we investigate the model's behavior in the local vicinity of the decision boundary between the forget and retain samples, where an unlearning process affects decision boundaries in a way that inadvertently degrades the representations of nearby retained data. To formalize this, we introduce an over-unlearning metric, OU $@\varepsilon$, which quantifies over-unlearning by measuring the changes in the model's predictive distribution after unlearning for samples adversarially perturbed within an $\varepsilon$-ball of the forget set. As a remedy, we propose a masked distillation loss to suppress the model output changes for the perturbed samples, which is further added to the existing MUs to relieve over-unlearning. To counter the PRA, we demonstrate that current unlearning methods fail to sufficiently collapse the latent class representations, leaving them susceptible to rapid reconstruction. Thus, we propose a dispersion loss to scatter the intra-class features of the forget data as widely as possible to neutralize the PRA, thereby enhancing robustness against PRA in conjunction with existing MU methods.

We evaluate across multiple datasets and architectures, including an identity-centric face recognition setting that mirrors practical privacy-driven unlearning requests. The observed trends are consistent across MU baselines: over-unlearning concentrates near forget-adjacent regions, and PRA remains a tangible post-unlearning threat.

Our contributions are summarized as follows:

- We present OU $@\varepsilon$, an efficient, retain-data-free metric capturing over-unlearning by measuring distributional drift near the decision boundary of the forget set.
- We raise a new threat called Prototypical Relearning Attack, which leverages residual feature structure to restore forgotten classes from only a handful of samples.
- We introduce Spotter, a plug-and-play class-level unlearning framework that simultaneously mitigates over-unlearning and neutralizes Prototypical Relearning Attacks by combining masked knowledge distillation around boundary-proximal perturbations and feature-dispersion objectives.

## 2. Related Works

**Machine Unlearning.** Machine unlearning (MU) aims to selectively remove the influence of target training data from learned models (Cao & Yang, 2015; Ginart et al., 2019). Early unlearning techniques focused on naively retraining from scratch by excluding forget data or approximating this process more efficiently. However, considering retraining

as a gold standard for unlearning classes/concepts is partially limited (Cooper et al., 2025), as it essentially aims to remove the sample-wise effects rather than eliminating conceptual knowledge. Moreover, regulations that mandate disposing of training data after model development can restrict access to the dataset. Consequently, prior work has explored class/concept-level unlearning via parameter masking/resetting (Thudi et al., 2022; Golatkar et al., 2020; Neel et al., 2021; Fan et al., 2024), decision-boundary shifting (Chen et al., 2023), data partitioning (Bourtoule et al., 2021; Hayase et al., 2020), and knowledge distillation (Chundawat et al., 2023; Kurmanji et al., 2023; Zhou et al., 2025). MU has also been extended to settings such as LLMs and federated learning (Ye et al., 2025; Liu et al., 2024; Lin et al., 2024; Liu et al., 2025; Wang et al., 2025b; Liu & Liu, 2025).

**Evaluations of MU and Over-Unlearning.** Evaluating MU is challenging (Thaker et al., 2025), as it must jointly achieve two closely coupled objectives: completely forgetting the target while preserving retained utility. Early studies often used coarse metrics, e.g., targeting a near-random accuracy on the forget class alongside pursuing high accuracies for the retained classes (Ali et al., 2025). For LLMs, various metrics have been proposed to evaluate the efficacy of MU processes tailored to specific language tasks (Shi et al., 2025; Lynch et al., 2024; Wang et al., 2025a).

Meanwhile, over-unlearning has been repeatedly reported as a practical concern (Hu et al., 2024a; Hayes et al., 2025), motivating attempts to mitigate over-unlearning (Ma et al., 2023; Wang et al., 2025a). However, in class-level unlearning, over-unlearning is still commonly assessed via global summaries on retained data or limited proxies (Hu et al., 2024a), which can obscure where the damage concentrates. Motivated by this gap, we propose an over-unlearning metric that focuses on the forget-adjacent region, capturing near-boundary degradation around the forget classes.

**Threats to Unlearning: Relearning Attack.** MU promises data removal, yet it raises new risks such as slow-down (Marchant et al., 2022), camouflaged poisoning (Di et al., 2023), inversion (Hu et al., 2024b), and relearning attacks (Hu et al., 2025). Specifically, a relearning attack refers to an adversary's attempt to restore high accuracy on the forgotten data by probing or fine-tuning the unlearned model with only a small subset of forget examples. In the context of LLMs, this threat has been emphasized by Lynch et al. (2024), who argue that fine-tuning on a small dataset can "jog" the model's memory and resurrect erased knowledge. To mitigate such attacks, a recent study suggests incorporating loss-landscape-aware objectives (e.g., sharpness-aware minimization, SAM (Foret et al., 2021)) into the unlearning process (Fan et al., 2025). Beyond LLMs, relearning attacks and follow-up defenses have begun to be explored in diffusion models (Gao et al., 2025; Yuan et al., 2025).

# 3. Blind Spots in Unlearning

In this section, we identify two key blind spots of MU, i.e., over-unlearning and relearning attack. First, we formalize a rigorous metric to quantify the unintended collateral damage inflicted on retained data during MU, denoted as OU @$\varepsilon$. Second, we demonstrate how the seemingly well-unlearned models can be compromised by a few-shot relearning attack, eventually leading to the recovery of forgotten data; we refer to this threat as the Prototypical Relearning Attack.

## 3.1. Formulation of Over-Unlearning

**Motivation: Over-unlearning is boundary-local.** A common way to assess retention after unlearning is to report average accuracy on retained classes. However, this global metric can hide where collateral damage occurs. In class-level unlearning, suppressing an entire class reshapes the decision boundary around the forget region, so the most vulnerable retained samples are likely those closest to the forget class. To examine this, we evaluate retained samples whose original model embeddings are nearest to the forget-class prototype. As shown in Appendix A, many methods preserve high overall retained accuracy but suffer larger drops on the top-2/5/10% forget-adjacent retained subsets. This motivates a boundary-proximal view of over-unlearning, where the key question is whether retained-class structure near the forget region remains intact after unlearning.

**Setup.** Let $\mathcal{D} = \mathcal{D}_r \cup \mathcal{D}_f$ be the original training set, where $\mathcal{D}_f$ denotes the *forget set* that must be erased and $\mathcal{D}_r$ the *retained set*. The training process yields a model parameterized by $\theta \leftarrow \text{TRAIN}(\mathcal{D})$. A MU procedure is performed by an algorithm $\mathcal{U}$ that is proposed to update $\theta$ to $\theta_u$ for the deletion request $\mathcal{D}_f$ without retraining on $\mathcal{D}_r$ from scratch. We focus on a supervised classification setting, where $\mathcal{D}$ is composed of $C$ different classes. As a starting point, let $f_\theta \colon \mathbb{R}^d \to \mathbb{R}^C$ be the *original* classifier trained on $\mathcal{D}$. We denote the *logit vector* of the classifier as $\boldsymbol{z}(\boldsymbol{x}; \theta)$, where $z_k(\boldsymbol{x}; \theta)$ represents the logit corresponding to class $k \in [C]$, and $[C] = \{1, \cdots, C\}$. To formulate the margin between the ground truth label $c \in [C]$ and others, we define the *logit-margin function*:

$$g_c(\boldsymbol{x}; \theta) := z_c(\boldsymbol{x}; \theta) - z_{c'}(\boldsymbol{x}; \theta),$$

where $c' = \arg\max_{k \neq c} z_k(\boldsymbol{x}; \theta)$.

**A Perturbed Set around Forget Data.** Let $\Delta_\varepsilon(\cdot; \theta)$ be a perturbation generator with budget $\varepsilon$. We define

$$\mathcal{A}_\varepsilon(\mathcal{D}_f; f_\theta) := \{\boldsymbol{x} + \delta \in \mathbb{R}^d \mid \boldsymbol{x} \in \mathcal{D}_f, \ \delta \in \Delta_\varepsilon(\boldsymbol{x}; \theta)\}. \tag{1}$$

For brevity, we denote $\mathcal{A}_\varepsilon(\mathcal{D}_f; f_\theta)$ as $\mathcal{A}_\varepsilon(\mathcal{D}_f)$. Intuitively, $\mathcal{A}_\varepsilon(\mathcal{D}_f)$ collects perturbed variants of the forget examples within an $\varepsilon$-neighborhood, and we may instantiate $\Delta_\varepsilon$ to reflect different notions of "vicinity." When emphasizing boundary-proximal perturbations, $\Delta_\varepsilon$ can be viewed as selecting perturbations that approach the decision boundary, e.g., $\Delta_\varepsilon(\boldsymbol{x}; \theta) \subseteq \{\delta \mid g_c(\boldsymbol{x} + \delta; \theta) < 0, \ \|\delta\| \leq \varepsilon\}$. Notably, the samples in $\mathcal{A}_\varepsilon(\mathcal{D}_f)$ reside in regions most directly influenced by the unlearning process. When we only require local perturbations around $x$, a natural choice is $\Delta_\varepsilon(\boldsymbol{x}; \theta) = \{\delta \mid \|\delta\| \leq \varepsilon\}$. We posit that over-unlearning manifests as the degradation in the model's retained data capability, with a particular focus on the samples in $\mathcal{A}_\varepsilon(\mathcal{D}_f)$ containing borderline data nearby $\mathcal{D}_f$.

**Over-unlearning Metric.** Based on such intuition, we opt to measure the degree of over-unlearning by calculating the divergence between the original model's masked probabilities and the unlearned model's probabilities, with a particular focus on the samples $\boldsymbol{x}_\mathrm{p} \in \mathcal{A}_\varepsilon(\mathcal{D}_f)$. Suppose that $\sigma(\boldsymbol{z})$ denotes the softmax function for $\boldsymbol{z} \in \mathbb{R}^C$ and let $\mathcal{C}_f \subseteq [C]$ be the set of forget classes. We define a masked softmax over $\mathcal{C}_f$ as follows:

$$\tilde{\sigma}(\boldsymbol{z}; \mathcal{C}_f) = \frac{1}{\sum_{c \notin \mathcal{C}_f} \sigma(\boldsymbol{z})_c} (\sigma(\boldsymbol{z}) \odot \boldsymbol{m}), \tag{2}$$

where $\boldsymbol{m} \in \{0, 1\}^C$ is a binary mask vector whose element $m_c = 0$ if $c \in \mathcal{C}_f$ and $m_c = 1$ otherwise. This masked softmax effectively renormalizes the probabilities to consider only the retained classes, preserving their relative likelihood. Thus, we define the over-unlearning metric OU @$\varepsilon$, which measures the collateral degradation of the unlearned model on $\mathcal{A}_\varepsilon(\mathcal{D}_f)$ as follows:

$$\text{OU} @\varepsilon := \mathbb{E}_{\boldsymbol{x}_\mathrm{p} \sim \mathcal{A}_\varepsilon(\mathcal{D}_f)} \big[ D\big( \tilde{\sigma}(\boldsymbol{z}(\boldsymbol{x}_\mathrm{p}; \theta)) \,\big\|\, \sigma(\boldsymbol{z}(\boldsymbol{x}_\mathrm{p}; \theta_u)) \big) \big], \tag{3}$$

where $D(\cdot \| \cdot)$ is a divergence measure between two probability vectors, such as Kullback-Leibler (KL) and Jensen-Shannon (JS) divergences. A larger OU @$\varepsilon$ indicates greater deviation of the unlearned model's prediction from the masked softmax output of the original model near the forget boundary, hence signifying more severe over-unlearning.

Notably, evaluating over-unlearning using retained data typically requires the laborious identification of non-forgotten samples that are closely entangled with the forget set and direct access to the retained dataset to measure the performance degradation. In cases where AI services are built upon publicly released foundation models, direct access to the underlying training data is infeasible. A key advantage of our metric OU @$\varepsilon$ is the feasibility of being computed solely from perturbed forget examples, eliminating any reliance on retained data and directly measuring the unintended forgetting. Intuitively, OU @$\varepsilon$ quantifies how well the unlearned model preserves the original model's knowledge around the decision boundary of the forget set, except for the information to be forgotten. For practical computation, we approximate the expectation over $\mathcal{A}_\varepsilon(\mathcal{D}_f)$

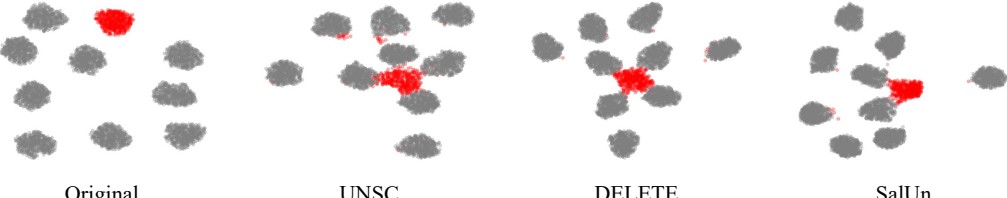

| Original | UNSC | DELETE | SalUn |

*Figure 1.* UMAP (McInnes et al., 2018) visualizations of CIFAR-10 representations computed with unlearned feature extractors, where the forget class is highlighted in ('red') and the remaining classes are shown in ('gray').

in Equation (3) using either (i) a **worst-case** attack such as Projected Gradient Descent (PGD) (Madry et al., 2018) that maximizes the loss of $f_\theta$, or (ii) a **random** perturbation such as Gaussian noise. Our primary results are based on PGD, while Gaussian noise results are provided in Appendix B to highlight the consistency of observed over-unlearning trends across different perturbation strategies.

### 3.2. Relearning Attack

**Motivation: Forgotten classes can remain clustered.** A *relearning attack* probes an unlearned model $f_{\theta_u}$ with a small subset of forget data in an attempt to re-induce the exact knowledge that the unlearning procedure sought to erase. In our toy examples, Figure 1 shows that state-of-the-art unlearning algorithms still produce feature spaces in which samples from the unlearned class ('red') remain clustered, despite their complete absence from the decision boundary. This implies that, even after the removal of an individual's identity information, the model retains sufficient capacity to distinguish or infer that identity. This clustering is often a byproduct of conventional unlearning methods that primarily focus on removing the influence of forget examples to maintain accuracy on retained data. Based on our observations, we hypothesize that, as in LLMs, relearning attacks with only a few forget samples could be effective. However, as shown in Figure 2-(a), existing fine-tuning-based relearning attacks (Lynch et al., 2024; Deeb & Roger, 2024; Hu et al., 2025) have not effectively exploited this clustered structure to recover the forget accuracy (denoted as $\text{Acc}_f$) in the image classification setting. While Boundary Shrink (Chen et al., 2023) shows some success cases, it was accompanied by a significant deterioration in the model's general utility. However, the failure of these relearning attacks does not imply safety; we introduce a novel attack methodology designed to leverage the underlying clusters.

**Prototypical Relearning Attack.** Let $\phi_\theta : \mathbb{R}^d \to \mathbb{R}^n$ denote the feature extractor of the classifier $f_\theta$. A prototypical network (Snell et al., 2017) computes a prototype for each class $c$ using its support set $S^{(c)} = \{\boldsymbol{x}_1^{(c)}, \ldots, \boldsymbol{x}_k^{(c)}\}$ as follows:

$$\mathbf{p}_\theta^{(c)} = \frac{1}{k} \sum_{i=1}^{k} \phi_\theta(\boldsymbol{x}_i^{(c)}) \in \mathbb{R}^n. \quad (4)$$

Given a query point $\boldsymbol{x}$, the predicted class is determined by

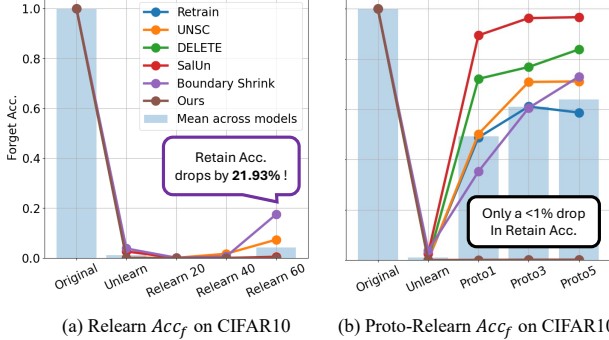

| (a) Relearn $Acc_f$ on CIFAR10 | (b) Proto-Relearn $Acc_f$ on CIFAR10 |

*Figure 2.* (a) $\text{Acc}_f$ comparisons of Original, Unlearned, and Relearned models. Unlearned models without relearning ('Unlearn'), and the relearned models on $N$ samples with a single epoch ('RelearnN'). (b) $\text{Acc}_f$ comparisons of Original, Unlearned, and Prototypical Relearned models. Only $N$ samples are used for computing the class-prototype ('ProtoN'). We set the hyperparameter $\alpha$ so that the drop in $\text{Acc}_r$ does not exceed 1%.

finding the prototype $\mathbf{p}_\theta^{(c)}$ that minimizes the distance $d(\cdot, \cdot)$ in the embedding space:

$$\arg\min_c d\left(\phi_\theta(\boldsymbol{x}), \mathbf{p}_\theta^{(c)}\right) = \arg\max_c \cos\left(\phi_\theta(\boldsymbol{x}), \mathbf{p}_\theta^{(c)}\right)$$
$$= \arg\max_c \phi_\theta(\boldsymbol{x})^\top \mathbf{p}_\theta^{(c)} / \|\mathbf{p}_\theta^{(c)}\| \quad (5)$$
$$= \arg\min_c \left\|\phi_\theta(\boldsymbol{x}) - \mathbf{p}_\theta^{(c)}\right\|^2$$
$$= \arg\max_c 2\phi_\theta(\boldsymbol{x})^\top \mathbf{p}_\theta^{(c)} - \left\|\mathbf{p}_\theta^{(c)}\right\|^2, \quad (6)$$

where $\|\phi_\theta(\boldsymbol{x})\|$ is constant with respect to the class $c$. Hence, this is equivalent to expressing the restored logit function $\hat{\boldsymbol{z}}(\boldsymbol{x}; \theta) = \hat{\mathbf{W}}\phi_\theta(\boldsymbol{x}) + \hat{b}$ as:

- Cosine similarity (5): $\hat{\mathbf{w}}_c = \mathbf{p}_\theta^{(c)} / \|\mathbf{p}_\theta^{(c)}\|, \quad \hat{b}_c = 0,$
- $L_2$ distance (6): $\hat{\mathbf{w}}_c = 2\mathbf{p}_\theta^{(c)}, \quad \hat{b}_c = -\|\mathbf{p}_\theta^{(c)}\|_2^2.$

Our PRA leverages this equivalence for efficient prediction of an unlearned model $\theta_u$ using only a few forget samples. Remarkably, it can even be effective with **a single sample** ('Proto1'), bypassing explicit prototype computation. Specifically, it replaces the weights and biases of the unlearned head $\boldsymbol{z}(\boldsymbol{x}, \theta_u) = \mathbf{W}^{(u)}\phi_{\theta_u}(\boldsymbol{x}) + b^{(u)}$ for the forget classes $c \in \mathcal{C}_f$ with the derived prototype weights $\hat{\mathbf{w}}_c$ and biases $\hat{b}_c$. Moreover, we interpolate these prototype weights

and biases with the corresponding parameters of the unlearned model to balance attack efficacy and generalization:

$$
\begin{aligned}
\mathbf{w}_c^\star &= \alpha\,\hat{\mathbf{w}}_c + (1-\alpha)\mathbf{w}_c^{(u)}, \\
b_c^\star &= \alpha\,\hat{b}_c + (1-\alpha)b_c^{(u)},
\end{aligned}
\tag{7}
$$

where the interpolation factor $\alpha \in [0,1]$ controls the trade-off. Figure 2-(b) demonstrates the effectiveness of our approach across various unlearning methods. We provide the algorithmic description of PRA and an ablation study on $\alpha$ in Appendix C.1 and Appendix C.2, respectively.

Our PRA suggests that, despite a service operator's full execution of an unlearning request and certification that the influence of the target data has been eliminated, a critical vulnerability remains: an adversary (often the service operator with *white-box* access) can rapidly restore the model's original performance on the forget data using only a handful of readily obtainable images. Crucially, this assumption is not merely hypothetical: in real-world vision deployments (e.g., face recognition), such seed images can be collected from social network services, enabling a lightweight post-hoc adaptation that can restore class-discriminative features that unlearning aimed to remove.

## 4. `Spotter`: An Unlearning Framework Against Over-Unlearning and Prototypical Relearning Attacks

Motivated by two key limitations in current unlearning practices, we propose a novel approach, `Spotter`, to address: (i) the lack of an explicit regularizer to mitigate over-unlearning, as formally quantified by OU@$\varepsilon$; and (ii) the susceptibility of post-unlearning representations to PRA.

**(i) Unlearning Objectives with Over-unlearning Mitigation.** Let $\mathcal{D}_f$ denote the forget set and $\mathcal{A}_\varepsilon(\mathcal{D}_f)$ its perturbed examples (as defined in Equation (1)). We present a soft unlearning loss based on a masked softmax function (as shown in Equation (2)) that aims to distill the distributional knowledge of the original model for $\mathcal{D}_f$ while ensuring that the probabilities for forget classes ($c \in \mathcal{C}_f$) vanish at zero (Hinton et al., 2015). To further mitigate over-unlearning, we apply the same loss to the perturbed set $\mathcal{A}_\varepsilon(\mathcal{D}_f)$ as a penalty, thereby regularizing the local decision boundary around the forget examples. The unlearning and over-unlearning losses are defined as:

$$
\mathcal{L}_u(\theta_u) = \frac{\sum_{\boldsymbol{x}\in\mathcal{D}_f} D\big(\tilde{\sigma}(\boldsymbol{z}(\boldsymbol{x};\theta))\,\|\,\sigma(\boldsymbol{z}(\boldsymbol{x};\theta_u))\big)}{|\mathcal{D}_f|},
\tag{8}
$$

$$
\mathcal{L}_o(\theta_u) = \frac{\sum_{\boldsymbol{x}_p\in\mathcal{A}_\varepsilon(\mathcal{D}_f)} D\big(\tilde{\sigma}(\boldsymbol{z}(\boldsymbol{x}_p;\theta))\,\|\,\sigma(\boldsymbol{z}(\boldsymbol{x}_p;\theta_u))\big)}{|\mathcal{A}_\varepsilon(\mathcal{D}_f)|}.
\tag{9}
$$

where $\tilde{\sigma}(\cdot)$ is a shorthand for the masked softmax output, i.e., $\tilde{\sigma}(\cdot;\mathcal{C}_f)$, of the teacher model $f_\theta$ (see Equation (2)),

and $D(\cdot,\cdot)$ is a generic divergence measure—instantiated as KL divergence in our case.

**(ii) Intra-class Dispersion Objective.** As shown in Figure 1, tightly clustered unlearned feature embeddings are susceptible to PRA. This vulnerability arises when the feature representations of forget samples are insufficiently dispersed, allowing attackers to easily exploit their structure and indicating incomplete unlearning. To address this, we introduce an Intra-class Dispersion Objective $\mathcal{L}_{sim}$, which computes the averaged pairwise cosine similarities between the samples of the unlearned feature embeddings for each forget class. This encourages the dispersion of these embeddings, thereby mitigating the risk of relearning attacks:

$$
\mathcal{L}_{sim}(\theta_u) = \sum_{c\in\mathcal{C}_f} \frac{\sum_{\boldsymbol{x}_i,\boldsymbol{x}_j\in\mathcal{D}_f^{(c)},\forall i\neq j} \frac{\phi_{\theta_u}(\boldsymbol{x}_i)^\top \phi_{\theta_u}(\boldsymbol{x}_j)}{\|\phi_{\theta_u}(\boldsymbol{x}_i)\|\|\phi_{\theta_u}(\boldsymbol{x}_j)\|}}{|\mathcal{C}_f|\cdot|\mathcal{D}_f^{(c)}|\cdot(|\mathcal{D}_f^{(c)}|-1)},
\tag{10}
$$

where $\mathcal{D}_f^{(c)}$ is the set of forget samples for class $c$. Minimizing $\mathcal{L}_{sim}$ disperses the feature embeddings of forget examples, making them less representative of prototypes.

**Overall Objective.** Our final objective function $\mathcal{L}$ combines three previously defined losses with weights $\lambda_1 \in [0,1]$ and $\lambda_2 \geq 0$. Specifically, $\lambda_1$ balances the standard unlearning loss $\mathcal{L}_u$ (8) and the over-unlearning mitigation loss $\mathcal{L}_o$ (9), while $\lambda_2$ controls the contribution of the intra-class dispersion loss $\mathcal{L}_{sim}$ (10):

$$
\mathcal{L} = \lambda_1\,\mathcal{L}_u + (1-\lambda_1)\,\mathcal{L}_o + \lambda_2\,\mathcal{L}_{sim}.
\tag{11}
$$

The joint optimization of $\mathcal{L}_u$ and $\mathcal{L}_o$ permits `Spotter` to achieve effective forgetting while mitigating the risk of over-unlearning, preserving the intrinsic structure of representations for the retained data. On the other hand, the intra-class dispersion term $\mathcal{L}_{sim}$ is essential in addressing this vulnerability against relearning attacks by explicitly encouraging the forget embeddings to spread out. Although $\mathcal{L}_{sim}$ can potentially influence the overall embedding space, the primary objective of $\mathcal{L}_u$ and $\mathcal{L}_o$ remains to minimize any disruption to the representations of retained data. Thus, this combined objective provides a comprehensive approach to tackle over-unlearning and PRA vulnerabilities.

Notably, our `Spotter`'s first term, $\mathcal{L}_u$, offers **plug-and-play compatibility**: it can be replaced with existing unlearning loss. `Spotter` enhances this base unlearning objective by incorporating $\mathcal{L}_o$ and $\mathcal{L}_{sim}$, thus enabling unlearning algorithms to address both over-unlearning and prototype vulnerabilities. This extension requires only minor modifications to the objective and introductions of two hyperparameters, $(\lambda_1, \lambda_2)$, without using any additional retained data.

*Table 1.* Comparisons of the unlearning performances among various methods. For CIFAR-10, we randomly unlearn one class, and for CIFAR-100, we randomly unlearn ten classes. Methods that utilize the retained data during the unlearning process are highlighted in ('blue'). We present the averaged results after conducting the experiment three times.

| Method | CIFAR-10 | | | | | | | CIFAR-100 | | | | | | |
|---|---|---|---|---|---|---|---|---|---|---|---|---|---|---|
| | $\mathrm{Acc}_f\downarrow$ | $\mathrm{Acc}_r\uparrow$ | $\mathrm{Acc}_{ft}\downarrow$ | $\mathrm{Acc}_{rt}\uparrow$ | OU @$\varepsilon\downarrow$ | Proto-Acc$_f\downarrow$ | $\mathrm{Acc}_r^*\uparrow$ | $\mathrm{Acc}_f\downarrow$ | $\mathrm{Acc}_r\uparrow$ | $\mathrm{Acc}_{ft}\downarrow$ | $\mathrm{Acc}_{rt}\uparrow$ | OU @$\varepsilon\downarrow$ | Proto-Acc$_f\downarrow$ | $\mathrm{Acc}_r^*\uparrow$ |
| Original Model | 100.00 | 100.00 | 94.70 | 93.98 | - | 100.00 | 100.00 | 99.98 | 99.98 | 74.90 | 75.76 | - | 100.00 | 99.99 |
| Retrain Model | 0.00 | 100.00 | 0.00 | 94.71 | 0.2384 | 58.70 | 99.86 | 0.00 | 99.99 | 0.00 | 76.31 | 0.2545 | 44.58 | 99.78 |
| Random Label | 0.84 | 99.50 | 0.70 | 91.41 | 0.1561 | 64.08 | 99.54 | 5.66 | 23.58 | 3.60 | 17.94 | 0.4450 | 30.18 | 18.70 |
| NegGrad | 0.18 | 87.73 | 0.30 | 81.37 | 0.3269 | 2.54 | 87.77 | 6.60 | 16.37 | 5.60 | 12.62 | 0.5309 | 10.98 | 15.61 |
| Boundary Shrink | 3.82 | 93.79 | 4.20 | 87.23 | 0.1435 | 72.96 | 92.87 | 7.06 | 12.86 | 6.40 | 11.19 | 0.4466 | 8.84 | 11.90 |
| Boundary Expand | **0.00** | 99.98 | **0.00** | 93.46 | 0.0958 | 99.98 | **99.98** | 8.70 | **99.98** | 4.90 | **76.31** | 0.0043 | 100.00 | 99.66 |
| SalUn | **0.00** | 98.18 | **0.00** | 89.09 | 0.1664 | 95.34 | 97.19 | 6.12 | 22.89 | 3.70 | 17.27 | 0.4481 | 11.70 | 22.55 |
| Learn to Unlearn | 0.62 | 97.43 | **0.00** | 89.36 | 0.3390 | 22.34 | 97.14 | 2.28 | 93.09 | 0.10 | 63.89 | 0.2397 | 11.80 | 92.19 |
| DELETE | **0.00** | 99.98 | **0.00** | **94.03** | 0.1216 | 83.84 | 99.07 | 0.12 | 98.77 | **0.00** | 73.49 | 0.2405 | 31.72 | 93.89 |
| Fisher | 0.12 | 93.99 | 0.10 | 87.04 | 0.1747 | 23.92 | 93.45 | **0.00** | 99.93 | **0.00** | 74.59 | 0.0491 | 96.74 | **99.91** |
| UNSC | **0.00** | 99.98 | **0.00** | 93.92 | 0.1575 | 71.10 | 99.75 | 0.62 | 99.79 | 3.10 | 73.89 | 0.1789 | 73.62 | 99.09 |
| **Spotter**($\lambda_2 = 0.1$) | **0.00** | **100.00** | **0.00** | **94.03** | **0.0139** | 62.12 | 99.89 | **0.00** | **99.98** | **0.00** | 76.16 | **0.0168** | 88.32 | 99.79 |
| **Spotter**($\lambda_2 = 1$) | **0.00** | 99.98 | **0.00** | 94.00 | 0.0228 | **0.24** | 99.96 | **0.00** | 99.96 | 0.20 | 75.94 | 0.0314 | **3.00** | 99.69 |

*Table 2.* Comparisons of the unlearning performances among various methods combined with **Spotter**. Cells of the form (before → after) show the value before and after applying **Spotter**. Forgetting performance and retention performance are comparable to prior methods or only marginally reduced.

| Method | CIFAR-10 | | | | | CIFAR-100 | | | | |
|---|---|---|---|---|---|---|---|---|---|---|
| | $\mathrm{Acc}_{ft}\downarrow$ | $\mathrm{Acc}_{rt}\uparrow$ | OU @$\varepsilon\downarrow$ | Proto-Acc$_f\downarrow$ | $\mathrm{Acc}_r^*\uparrow$ | $\mathrm{Acc}_{ft}\downarrow$ | $\mathrm{Acc}_{rt}\uparrow$ | OU @$\varepsilon\downarrow$ | Proto-Acc$_f\downarrow$ | $\mathrm{Acc}_r^*\uparrow$ |
| SalUn + **Spotter** | 0.00 | 93.37 | 0.1664 → **0.0345** | 95.34 → **0.34** | 98.07 | 3.20 | 48.32 | 0.4481 → **0.0964** | 11.70 → **4.44** | 63.00 |
| DELETE + **Spotter** | 0.00 | 92.64 | 0.1216 → **0.0232** | 83.84 → **0.04** | 99.37 | 0.90 | 73.99 | 0.2405 → **0.0429** | 31.72 → **3.34** | 99.38 |
| UNSC + **Spotter** | 0.00 | 93.43 | 0.1575 → **0.0089** | 71.10 → **0.82** | 99.98 | 4.00 | 73.00 | 0.1789 → **0.0418** | 73.62 → **18.54** | 99.38 |

## 5. Experiments

### 5.1. Experimental Setup

**Data and Models.** Main experiments are conducted on the CIFAR-10 and CIFAR-100 image classification benchmarks (Krizhevsky & Hinton, 2009). We train a ResNet-18 (He et al., 2016) on each dataset, using the full training split as the retained data $\mathcal{D}_r$ plus a forget subset $\mathcal{D}_f$. For CIFAR-10, we randomly sample $|\mathcal{C}_f| = 1$ class as the target of unlearning, whereas for CIFAR-100 we randomly sample $|\mathcal{C}_f| = 10$ classes. In Appendix E and F, we further evaluate on a larger-scale classification benchmark (Tiny-ImageNet) and a real-world face recognition dataset (CASIA-WebFace), and also consider a larger Transformer-based backbone.

**Baselines and Evaluations.** We examine 9 unlearning approaches, with detailed descriptions of each provided in the Appendix D. We report classification performance on the forget classes—training accuracy $\mathrm{Acc}_f$ and test accuracy $\mathrm{Acc}_{ft}$—and likewise on the retained classes—training accuracy $\mathrm{Acc}_r$ and test accuracy $\mathrm{Acc}_{rt}$. To quantify unintended over-unlearning effect, we compute the proposed

over-unlearning metric OU @$\varepsilon$ with $\varepsilon = 0.03$, using the JS divergence as the divergence measure. Specifically, we employ $\ell_\infty$ PGD attack with a random start against the original model, where perturbation budget $\varepsilon = 0.03$, step size $\eta = 0.01$, and 3 iterations to generate perturbed samples.

**Experiment Details of Spotter.** We apply the $\ell_\infty$ PGD attack to generate perturbed samples for use with Spotter. These perturbations are generated independently of the perturbed samples during the evaluation. We configured $\lambda_1 = 0.7$ and used $\lambda_2 = 0.1$ and 1.

**Prototypical Relearning Attack.** For every forget class, we randomly draw $k = 5$ query images, extract the pre-head embeddings from the unlearned ResNet-18, and form the prototype following Equation (4). We used the cosine similarity to derive prototype weights. The resulting accuracy on $\mathcal{D}_f$ is Proto-Acc$_f$, while the accuracy on $\mathcal{D}_r$ after the attack is $\mathrm{Acc}_r^*$. We tune the interpolation factor $\alpha$ and keep values for which the retained accuracy drop satisfies $\mathrm{Acc}_r^* \geq \mathrm{Acc}_r - 1\%$. This indicates a scenario in which the model owner fails to detect the attack because the model's utility shows little to no change.

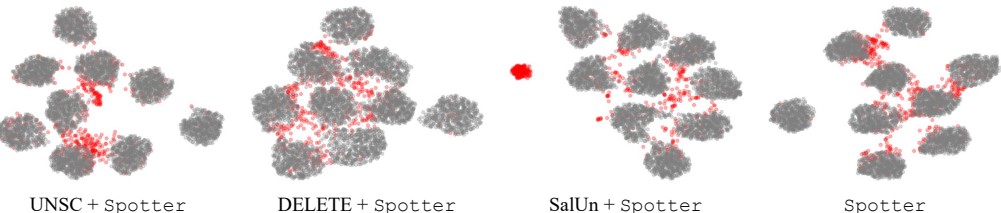

*Figure 3.* UMAP visualizations on CIFAR-10 representations computed with a `Spotter`-unlearned feature extractor, where the forget class is highlighted in ('red') and the remaining classes, ('gray').

## 5.2. Results

Table 1 compares unlearning methods in terms of forget efficacy ($Acc_{f,ft}$), utility retention ($Acc_{r,rt}$), over-unlearning ($OU @\varepsilon$), and prototypical relearning robustness (Proto-$Acc_f$, $Acc_r^*$). Table 2 shows `Spotter`'s compatibility by combining it with 3 representative methods.

**Unlearning Performance Comparison.** Random Label retains accuracy on CIFAR-10, but both over-unlearns and collapses on CIFAR-100. Boundary Expand strongly suppresses $Acc_f$ and yields a low $OU @\varepsilon$, yet is almost completely reversed by our PRA. Although it shifts the forget-class decision boundary via an auxiliary shadow class, the residual feature geometry remains exploitable for PRA. UNSC and DELETE preserve $Acc_r (> 99 \%)$ but still leak substantial prototype information and incur appreciable collateral damage. Prior techniques may achieve unlearning, but they frequently do so at the cost of over-unlearning and worsened susceptibility to the PRA. `Spotter` achieves *state-of-the-art (SotA)* performance on nearly all metrics when $\lambda_2 = 0.1$. Furthermore, setting $\lambda_2 = 1$ to enhance robustness against the PRA preserves near-SotA unlearning performance while delivering the strongest defense. These results indicate that `Spotter` delivers near-perfect class-level unlearning and mitigates over-unlearning, while also enabling proactive control of robustness against relearning attacks via hyperparameter tuning. Importantly, our findings generalize beyond the CIFAR/ResNet settings: `Spotter` remains effective on larger and real-world datasets, even when forgetting up to 500 classes (Appendix E) and on Transformer-based classifiers (Appendix F).

**`Spotter` reliably mitigates both blind spots.** When `Spotter` is applied to baselines, it caps over-unlearning, e.g., UNSC's $OU @\varepsilon$ falls from $0.178 \rightarrow 0.009$ ($\times 0.05$ reduction) and neutralizes PRA. Additionally, `Spotter` partially recovers retention on the collapsed baseline (SalUn + `Spotter`) despite not using retained data during unlearning, thus emulating the benefit that is commonly obtained from leveraging retained examples. This underscores a principal advantage of `Spotter`, namely its ability to erase the forget data while largely preserving the original decision boundary. `Spotter` and `Spotter` + baselines simultaneously achieve near-complete forgetting, preserve all perfor-

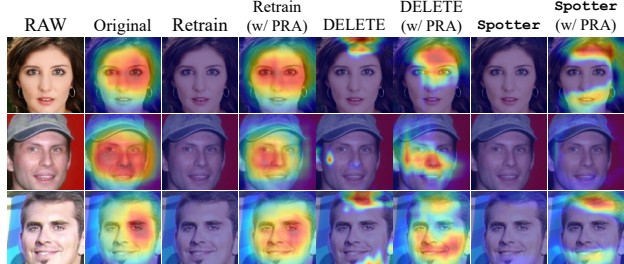

*Figure 4.* **Grad-CAM analysis on face recognition under PRA.** Class activation maps for forget-class face images across Original, Retrain, DELETE, and `Spotter`, clean and under the PRA.

mance on retained data, drive collateral damage down, and leave almost no room for prototypical relearning. Overall, attaching `Spotter` to existing pipelines yields results that help bring them closer to robust unlearning.

**Embedding Dispersion.** We visualize UMAP projections of CIFAR-10 embeddings in Figure 3 for `Spotter` and representative baselines combined with `Spotter`. Across methods, forget-class samples ('red') no longer form the tight, separable cluster seen in Figure 1, but dispersed over the manifold, while retained classes ('gray') remain compact. However, we note SalUn + `Spotter` leaves a small forget data cluster on the left edge of the manifold, suggesting a potential weakness: if a prototype were computed from that pocket, an adversary could partially rebuild the forgotten boundary. In Table 2, Proto-$Acc_f$ is nearly 0%. This is because PRA samples $n$ forget examples (but only a very small number of samples are allowed) uniformly at random. In our experiments, we calculate prototypes with only 5 forget samples. We want to emphasize that the attacker cannot know the overall feature distribution of the forget class, so it is not possible to intentionally draw a sample in a small forget cluster. To assess robustness, we also evaluate prototypes computed from larger query sets in Appendix G.

## 5.3. Various Unlearning Scenarios

**Face Recognition Task.** Figure 4 extends to an identity-centric setting by evaluating on a face recognition task. With Grad-CAM (Selvaraju et al., 2017), we examine not only whether a model "forgets" a target identity, but also what evidence it relies on before and after unlearning. These results

*Table 3.* Class-level unlearning performance on pretrained CLIP ViT-B/32 in an ImageNet classification setting.

| Method | $\text{Acc}_{ft}\downarrow$ | $\text{Acc}_{rt}\uparrow$ | $\text{OU}@\varepsilon\downarrow$ |
|---|---|---|---|
| Original Model | 54.18 | 55.75 | – |
| **Spotter** | **0.00** | 55.00 | 0.0141 |

highlight PRA's practical threat. For the strong baseline DELETE, PRA substantially increases the forget accuracy, and the attention maps re-concentrate on identity-critical facial landmarks (eyes, nose, and mouth), suggesting that a small amount of post-unlearning adaptation restores discriminative cues for the forgotten identity. In contrast, Spotter shows higher resistance: although PRA induces some activation around the face region, it remains comparatively weak on the most informative identity cues and tends to appear on forehead and surrounding regions. This qualitative pattern aligns with the limited recovery of forget accuracy under PRA, indicating that Spotter more effectively prevents the attack from reinstating identity-specific evidence.

**Extension to a Pretrained Vision-Language Model.** We further examine whether Spotter can be applied beyond a standard classifier with a learned linear head. To this end, we evaluate Spotter on a pretrained CLIP (Radford et al., 2021) ViT-B/32 in an ImageNet classification setting. Unlike the ResNet classifiers used in the main experiments, CLIP predicts a class by comparing an image embedding with fixed text-prompt embeddings. During unlearning, we keep the text encoder and text embeddings fixed and update only the visual encoder. The forget-class masking in Spotter is then applied directly to the CLIP class-probability vector: the teacher distribution is obtained by masking out the forget classes from the original CLIP logits and renormalizing the remaining class probabilities, while the unlearned visual encoder is trained to match this masked retained-class distribution. Thus, the masked distillation terms $\mathcal{L}_u$ and $\mathcal{L}_o$ suppress probability mass assigned to the forget classes without changing the text-side classifier prototypes. For the dispersion term $\mathcal{L}_{\text{sim}}$, we use the CLIP image embeddings of forget-class samples and minimize their intra-class cosine similarity, so that the visual representations of the removed classes no longer form compact prototype-like clusters. We randomly select 10 ImageNet classes as forget classes and compute $\text{OU}@\varepsilon$ using the same masked-distribution definition as in the main experiments, with PGD perturbations generated from the CLIP zero-shot classification loss. As shown in Table 3, Spotter reduces the forget-class test accuracy to $0.00\%$ while preserving retained-class accuracy with only a small drop from $55.75\%$ to $55.00\%$. The resulting $\text{OU}@\varepsilon$ remains low, suggesting that the proposed masked-distillation and dispersion objectives can also operate in a pretrained vision-language classifier, where class decisions are induced by fixed text prompts rather than a learned classifier head.

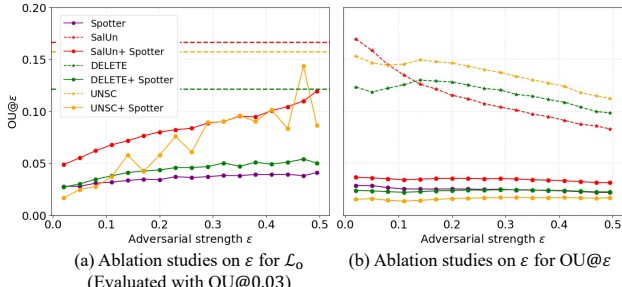

(a) Ablation studies on $\varepsilon$ for $\mathcal{L}_o$ (Evaluated with OU@0.03)  (b) Ablation studies on $\varepsilon$ for OU@$\varepsilon$

*Figure 5.* (a) Ablation studies on $\varepsilon$ for $\mathcal{L}_o$. We unlearn models with varying $\varepsilon$ used in $\mathcal{L}_o$ and measure the over-unlearning with $\varepsilon = 0.03$ perturbed set. (b) Ablation studies on $\varepsilon$ for $\text{OU}@\varepsilon$. We measure $\text{OU}@\varepsilon$ varying $\varepsilon$ used to generate the perturbed set. Dotted lines represent the case where Spotter is not applied.

**Challenging Unlearning Regimes.** We further evaluate Spotter in two practical regimes: sequential unlearning (10 rounds, 1 class/round) and severe class-level unlearning (30%/50% classes) on CIFAR-100. Across both settings, Spotter remains effective: it consistently forgets the target classes without degrading retained performance, while keeping collateral damage and prototype leakage small. Detailed results are reported in Appendix H and Appendix I. In practice, these match "unlearning-as-a-service" deployments, where deletion requests arrive continuously and the system must remain reliable without repeated retraining.

### 5.4. Ablation Study

$\varepsilon$ **on Spotter Unlearning.** We conduct unlearning experiments on SalUn, DELETE, and UNSC, as well as their Spotter-combined variants (including standalone Spotter), with PGD-perturbation radius values of $\varepsilon \in \{0.01, ..., .0.5\}$ for calculating $\mathcal{L}_o$. For over-unlearning evaluation, we utilize the same perturbed evaluation set ($\varepsilon = 0.03$) with the main experiments. The results are shown in Figure 5-(a). Overall, the over-unlearning metric rises as $\varepsilon$ increases. When $\varepsilon$ becomes too large, the perturbed samples deviate substantially from the original forget data and move away from the decision-boundary neighborhood that defines the target unlearning region, thus yielding less benefit. Nevertheless, it remained more robust to overunlearning than the baseline without Spotter.

$\varepsilon$ **on** $\text{OU}@\varepsilon$ **Evaluation.** Using the unlearned checkpoints reported in Table 2, we further evaluate the $\text{OU}@\varepsilon$ values for every test radius $\varepsilon \in \{0.01, ..., 0.5\}$ via PGD. From Figure 5-(b), collateral damage for baselines diminishes linearly with increasing $\varepsilon$, whereas Spotter and Spotter combined methods remain consistently low throughout the entire $\varepsilon$ range, confirming the reliability of $\text{OU}@\varepsilon$ as a metric for quantifying the collateral damage. The reason behind the decreasing behavior is that beyond some point, a larger radius pushes perturbation points too far from the decision boundaries, thereby the perturbed samples enter the internal region of the retained set, which shows a small $\text{OU}@\varepsilon$.

*Table 4.* Ablation studies on Spotter regularization coefficients $\lambda_1$ and $\lambda_2$ for CIFAR-10.

| Method | $\lambda_2 = 0$ | | | | $\lambda_2 = 1$ | | | | $\lambda_2 = 2$ | | | |
|---|---|---|---|---|---|---|---|---|---|---|---|---|
| | $Acc_{ft}\downarrow$ | $Acc_{rt}\uparrow$ | OU$@\varepsilon\downarrow$ | Proto-Acc$_f\downarrow$ | $Acc_{ft}\downarrow$ | $Acc_{rt}\uparrow$ | OU$@\varepsilon\downarrow$ | Proto-Acc$_f\downarrow$ | $Acc_{ft}\downarrow$ | $Acc_{rt}\uparrow$ | OU$@\varepsilon\downarrow$ | Proto-Acc$_f\downarrow$ |
| SalUn + **Spotter**($\lambda_1 = 1$) | 0.00 | 89.09 | 0.1864 | 95.34 | 0.00 | 86.18 | 0.2321 | 0.52 | 0.00 | 83.08 | 0.2329 | 0.06 |
| SalUn + **Spotter**($\lambda_1 = 0.7$) | 0.00 | 93.09 | 0.0335 | 95.42 | 0.00 | 90.13 | 0.0347 | 0.34 | 0.00 | 90.22 | 0.1025 | 0.18 |
| SalUn + **Spotter**($\lambda_1 = 0.3$) | 0.00 | 93.84 | 0.0222 | 91.50 | 0.00 | 93.37 | 0.0345 | 0.34 | 0.00 | 92.42 | 0.0346 | 0.04 |
| **Spotter**($\lambda_1 = 1$) | 0.00 | 94.22 | 0.0115 | 99.92 | 0.00 | 93.71 | 0.0485 | 0.76 | 0.00 | 91.93 | 0.1466 | 0.26 |
| **Spotter**($\lambda_1 = 0.7$) | 0.00 | 94.19 | 0.0094 | 99.70 | 0.00 | 93.91 | 0.0298 | 0.26 | 0.00 | 93.36 | 0.0433 | 0.18 |
| **Spotter**($\lambda_1 = 0.3$) | 0.00 | 94.18 | 0.0095 | 99.30 | 0.00 | 93.88 | 0.0197 | 0.24 | 0.00 | 93.79 | 0.0368 | 0.18 |

$\lambda_1$ **and** $\lambda_2$. Table 4 systematically explores the two Spotter hyperparameters $\lambda_1$ and $\lambda_2$ for SalUn + Spotter and Spotter. When $\lambda_1 = 1$ (without $\mathcal{L}_o$), Spotter behaves like a conventional unlearning objective: forgetting is perfect, but collateral damage remains high. Incorporating $\mathcal{L}_o$ mitigates the collateral damage while leaving retention intact. Orthogonally, increasing $\lambda_2$ enlarges the forget-class spread and therefore suppresses Proto-Acc$_f$ from near-total recovery to approximately zero, but at the cost of a few points of $Acc_{rt}$. A balanced configuration such as $\lambda_1 = 0.7$ and $\lambda_2 = 1$ achieves near-optimal performance. Crucially, these gains emerge only when *both* $\mathcal{L}_o$ and $\mathcal{L}_{sim}$ are active; disabling either loss immediately re-darkens one of the blind spots, underscoring that the two terms are complementary and must be deployed together.

## 6. Discussion

**Toward Robust Safety-Driven Unlearning.** Classical machine unlearning is usually grounded in deletion fidelity: after a request to remove a forget set, the updated model is expected to match, or be statistically indistinguishable from, a model trained without those data (Cao & Yang, 2015; Ginart et al., 2019; Bourtoule et al., 2021). This reference remains essential when the objective is certified removal of specific training records. However, recent work on generative AI has made clear that many deployed unlearning requests are also behavioral and safety-driven (Cooper et al., 2025). They ask the model not to reveal private content, reproduce copyrighted material, generate harmful content, or reacquire removed capabilities after further interaction or fine-tuning. Our work follows a practical unlearning objective of a similar kind. Rather than requiring strict retrain equivalence (Guo et al., 2020), we ask whether a class-level unlearning method can prevent the removed class or identity from being reliably recognized or rapidly recovered, while preserving the original model's retained knowledge as much as possible. Under this objective, the retrained model is not the normative reference. What matters is not only whether the influence of a finite forget set has been removed, but also whether residual class-level structure remains exploitable after unlearning. This perspective has recently gained attention in generative AI unlearning (Zhang et al., 2024; Fan et al., 2025; Zhang et al., 2025), where constructing an exact retraining reference is often infeasible, but it remains comparatively underexplored in classification models. Our Prototypical Relearning Attack highlights why this objective is also necessary in class-level unlearning: even when a model appears to have forgotten a target class under conventional accuracy-based evaluation, a small number of forget-class examples can be sufficient to recover its decision ability.

**Limitations and Future Work.** Here, we discuss the limitations and directions for future work. First, our formulation focuses on class-level unlearning, where the forget target is represented as labeled classes or identities. While this setting covers practical cases such as identity removal in face recognition, it does not directly address arbitrary subset deletion or attribute-level removal. Second, Spotter provides an empirical mitigation for boundary-proximal collateral damage and PRA, but it does not provide formal guarantees of removal or robustness. Finally, PRA is formulated as a white-box audit attack. Although it exposes residual class-level structure after unlearning, it does not cover limited-access or adaptive adversaries, leaving gray-box and black-box relearning attacks for future investigation.

## 7. Conclusion

We identified two overlooked blind spots in class-level unlearning: over-unlearning and relearning attacks. To quantify the former without retained data, we proposed OU$@\varepsilon$, which measures distributional drift near the forget decision boundary. We also introduced the Prototypical Relearning Attack, showing that a few forget samples rapidly restore forgotten performance by exploiting residual feature structure. To address both issues, we proposed Spotter, a novel unlearning framework that combines masked distillation on boundary-proximal perturbations with an intra-class dispersion loss to suppress prototype leakage, offering plug-and-play compatibility with existing unlearning methods. Across a wide range of datasets and model architectures, Spotter consistently reduces OU$@\varepsilon$ and neutralizes PRA while preserving utility. Our work illuminates critical blind spots in class-level unlearning and provides a remedy that advances machine unlearning toward reliable, regulation-conscious, and attack-resilient operation in AI systems.

# Acknowledgements

This work was supported by the National Research Foundation of Korea (NRF) grant funded by the Korea government (MSIT) (No. RS-2024-00459023), the Institute of Information & Communications Technology Planning & Evaluation (IITP) grant funded by the Korean Government (MSIT): (No. RS-2020-II201336, Artificial Intelligence Graduate School Program at UNIST), (No. RS-2025-25442824, AI Star Fellowship Program (UNIST)), (No. IITP-2026-RS-2022-00156361, Innovative Human Resource Development for Local Intellectualization Program), (No. RS-2026-25528781, Hyper-scale Industrial AI Research Support (R&D) Program, Development of an industry-specified intelligent data processing and federated learning platform), (No. IITP-2026-RS-2024-00436936, the Information Technology Research Center (ITRC)).

# Impact Statement

This paper audits the robustness of machine unlearning by introducing metrics for boundary-proximal over-unlearning and by analyzing post-hoc relearning threats. We propose a mitigation objective that improves utility on non-forgotten data and reduces relearning vulnerability without requiring retained data. The main benefit is more trustworthy unlearning for privacy and safety purposes. As a dual-use risk exists, we present the threat model to support auditing and accompany it with countermeasures.

# Software and Data

We publicly release the source code for reproducing our experiments, including the implementation of OU@$\epsilon$, the Prototypical Relearning Attack, and Spotter, at `https://github.com/Seung-B/Spotter-Unlearning`. The repository includes training and unlearning scripts, evaluation code, and instructions. All datasets used in this work are publicly available benchmarks.

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

## A. Validation of $\mathrm{OU}\,@\varepsilon$ with Forget-Adjacent Retained Samples

*Table 5.* Comparison of unlearning performance on CIFAR-10 when one class is randomly unlearned. In addition to Table 1, we report the accuracy on the retained test samples that are closest to the forget prototype in the representation space (e.g., Top-2/5/10%). Methods using retained data during unlearning are highlighted in blue. All results are averaged over three runs.

| Method | CIFAR-10 | | | | | | | |
|---|---|---|---|---|---|---|---|---|
| | $\mathrm{Acc}_f\downarrow$ | $\mathrm{Acc}_r\uparrow$ | $\mathrm{Acc}_{ft}\downarrow$ | $\mathrm{Acc}_{rt}\uparrow$ | Top 2% $\uparrow$ | Top 5% $\uparrow$ | Top 10% $\uparrow$ | $\mathrm{OU}\,@\varepsilon\downarrow$ |
| Original Model | 100.00 | 100.00 | 94.70 | 93.98 | 57.78 | 71.78 | 77.56 | - |
| Retrain Model | 0.00 | 100.00 | 0.00 | 94.71 | 81.11 | 81.56 | 84.11 | - |
| Random Label | 0.84 | 99.50 | 0.70 | 91.41 | 61.67 | 67.56 | 71.33 | 0.1561 |
| NegGrad | 0.18 | 87.73 | 0.30 | 81.37 | 37.78 | 47.56 | 54.89 | 0.3269 |
| Boundary Shrink | 3.82 | 93.79 | 4.20 | 87.23 | 54.44 | 60.89 | 66.33 | 0.1435 |
| Boundary Expand | **0.00** | 99.98 | **0.00** | 93.46 | 75.56 | 77.11 | 78.67 | 0.0958 |
| SalUn | **0.00** | 98.18 | **0.00** | 89.09 | 63.89 | 69.33 | 73.22 | 0.1664 |
| Learn to Unlearn | 0.62 | 97.43 | **0.00** | 89.36 | 53.89 | 57.11 | 64.11 | 0.3390 |
| DELETE | **0.00** | 99.98 | **0.00** | **94.03** | 74.44 | 76.67 | 78.78 | 0.1216 |
| Fisher | 0.12 | 93.99 | 0.10 | 87.04 | 56.11 | 60.67 | 65.11 | 0.1747 |
| UNSC | **0.00** | 99.98 | **0.00** | 93.92 | 75.56 | 77.33 | 79.56 | 0.1575 |
| `Spotter` | **0.00** | 99.98 | **0.00** | 94.00 | **77.22** | **78.67** | **80.56** | **0.0228** |

Our proposed $\mathrm{OU}\,@\varepsilon$ metric is intended to capture boundary-proximal collateral distortion, rather than global retained accuracy or retrain-equivalence. To further examine whether this metric reflects actual degradation on retained samples near the forget region, we conduct an auxiliary evaluation on forget-adjacent retained test samples. Specifically, for each unlearning run, we compute the prototype of the forget class in the representation space and rank retained test samples by their proximity to this prototype. We then evaluate retained test accuracy on the closest Top-2%, Top-5%, and Top-10% subsets. Table 5 shows that global retained accuracy can obscure local collateral damage. For example, several methods maintain high overall retained test accuracy, but their accuracy drops noticeably on retained samples closest to the forget prototype. This indicates that the effect of class-level unlearning is not uniformly distributed over the retained classes, but is concentrated around forget-adjacent regions. In contrast, `Spotter` achieves the lowest $\mathrm{OU}\,@\varepsilon$ and consistently preserves the highest accuracy on the Top-2%, Top-5%, and Top-10% retained subsets. These results support the interpretation of $\mathrm{OU}\,@\varepsilon$ as a retain-data-free proxy for boundary-proximal collateral distortion.

## B. Gaussian Noise based $\mathrm{OU}\,@\varepsilon$ Evaluation

*Table 6.* Comparison of PGD/Gaussian-based $\mathrm{OU}\,@\varepsilon$ on CIFAR-10/CIFAR-100 using unlearned models in Table 1.

| Method | CIFAR-10 | | CIFAR-100 | |
|---|---|---|---|---|
| | $\mathrm{OU}\,@\varepsilon\downarrow$ | Gaussian-$\mathrm{OU}\,@\varepsilon\downarrow$ | $\mathrm{OU}\,@\varepsilon\downarrow$ | Gaussian-$\mathrm{OU}\,@\varepsilon\downarrow$ |
| Random Label | 0.1561 | 0.2041 | 0.4450 | 0.3747 |
| NegGrad | 0.3269 | 0.3161 | 0.5309 | 0.5174 |
| Boundary Shrink | 0.1435 | 0.1569 | 0.4466 | 0.4176 |
| Boundary Expand | 0.0958 | 0.0914 | 0.0043 | 0.0043 |
| SalUn | 0.1664 | 0.1946 | 0.4481 | 0.3791 |
| Learn to Unlearn | 0.3390 | 0.3460 | 0.2397 | 0.2815 |
| DELETE | 0.1216 | 0.1036 | 0.2405 | 0.2982 |
| Fisher | 0.1747 | 0.1749 | 0.0491 | 0.0541 |
| UNSC | 0.1575 | 0.1275 | 0.1789 | 0.2363 |
| `Spotter` | 0.0228 | 0.0272 | 0.0314 | 0.0322 |

In addition to the PGD-based worst-case perturbations described in Section 3.1, we assess the robustness of the OU @$\varepsilon$ metric under random perturbations sampled from a Gaussian distribution. Specifically, for each forget sample $x \in \mathcal{D}_f$, we draw a noise $\delta \sim \mathcal{N}(0, \sigma^2 I)$ with variance $\sigma^2 = 0.01$ to form perturbed sample $x + \delta$. We then compute the over-unlearning score on these Gaussian-perturbed samples—denoted Gaussian-OU @$\varepsilon$—in the same fashion as OU @$\varepsilon$ (Eq. 3).

Table 6 compares PGD-based OU @$\varepsilon$ and Gaussian-OU @$\varepsilon$ across all unlearning methods on CIFAR-10 and CIFAR-100. The results demonstrate that both perturbation strategies yield highly consistent trends: methods that exhibit high OU @$\varepsilon$ under PGD also show high Gaussian-OU @$\varepsilon$, and vice versa. However, while Gaussian noise confirms the stability of OU @$\varepsilon$ as a metric, we adopt PGD for the main experiments to more precisely probe boundary-proximal over-unlearning.

## C. Details of Prototypical Relearning Attack

### C.1. Algorithmic Description

---

**Algorithm 1:** Prototypical Relearning Attack

---

**Input:** Unlearned model $f_{\theta_u} = (\phi_{\theta_u}, z_{\theta_u})$; small forget set $S_f = \{(\boldsymbol{x}_i, y_i)\}_{i=1}^N$; samples per class $k$; interpolation factor
$\alpha \in [0, 1]$; distance metric $d \in \{\cos, \ell_2\}$

1   Let $\mathcal{C}_f = \{ c : \exists (\boldsymbol{x}, y) \in S_f, \ y = c \}$
2   **foreach** $c \in \mathcal{C}_f$ **do**
3     Let $S_f^{(c)} = \{ \boldsymbol{x} : (\boldsymbol{x}, y) \in S_f, \ y = c \}$
4     Compute prototype: $\mathbf{p}_c \leftarrow \frac{1}{|S_f^{(c)}|} \sum_{\boldsymbol{x} \in S_f^{(c)}} \phi_{\theta_u}(\boldsymbol{x})$
5     **if** $d = \cos$ **then**
6       $\hat{\mathbf{w}}_c \leftarrow \mathbf{p}_c / \|\mathbf{p}_c\|, \quad \hat{b}_c \leftarrow 0$
7     **else**
8       $\hat{\mathbf{w}}_c \leftarrow 2\,\mathbf{p}_c, \quad \hat{b}_c \leftarrow -\|\mathbf{p}_c\|_2^2$
9     Interpolate with original(unlearned) head parameters $\mathbf{w}_c^{(u)}$: $\mathbf{w}_c^\star \leftarrow \alpha\,\hat{\mathbf{w}}_c \ + \ (1 - \alpha)\,\mathbf{w}_c^{(u)}, \quad \mathbf{b}_c^\star \leftarrow \alpha\,\hat{b}_c \ + \ (1 - \alpha)\,b_c^{(u)}$
10   Form patched unlearned head $\boldsymbol{z}^\star(\boldsymbol{x}; \theta_u) = W^\star \phi_{\theta_u}(\boldsymbol{x}) + b^\star$ by replacing only the rows for classes in $\mathcal{C}_f$ in $(W^{(u)}, b^{(u)})$
11   **return** $\boldsymbol{z}^\star(x; \theta_u)$

---

### C.2. Ablation Study on $\alpha$

*Table 7.* Ablation study on interpolation factor $\alpha$.

| Method | $\alpha = 1$ | | $\alpha = 0.8$ | | $\alpha = 0.6$ | | $\alpha = 0.4$ | | $\alpha = 0.2$ | |
|---|---|---|---|---|---|---|---|---|---|---|
| | Proto-Acc$_f\downarrow$ | Acc$_r^*\uparrow$ | Proto-Acc$_f\downarrow$ | Acc$_r^*\uparrow$ | Proto-Acc$_f\downarrow$ | Acc$_r^*\uparrow$ | Proto-Acc$_f\downarrow$ | Acc$_r^*\uparrow$ | Proto-Acc$_f\downarrow$ | Acc$_r^*\uparrow$ |
| Retrain | 58.70 | 99.86 | 27.40 | 100.00 | 2.24 | 100.00 | 0.00 | 100.00 | 0.00 | 100.00 |
| UNSC | 74.78 | 99.69 | 45.62 | 99.98 | 15.70 | 100.00 | 0.60 | 100.00 | 0.00 | 100.00 |
| DELETE | 90.92 | 97.40 | 70.76 | 99.80 | 29.28 | 99.99 | 3.66 | 100.00 | 0.22 | 100.00 |
| SalUn | 97.42 | 97.84 | 94.70 | 99.35 | 76.18 | 99.71 | 36.08 | 99.73 | 9.76 | 99.81 |
| Boundary Shrink | 90.62 | 84.49 | 74.72 | 93.13 | 42.90 | 93.78 | 16.08 | 93.81 | 7.64 | 93.88 |
| Spotter | **0.22** | 99.96 | **0.26** | 99.96 | **0.12** | 99.96 | **0.06** | 99.96 | **0.06** | 99.96 |

Table 7 reports how the Prototypical Relearning Attack responds to different values of the interpolation factor $\alpha$, which linearly blends prototype logits with the original unlearned head (Eq. 7). Recall that Proto-Acc$_f$ is the accuracy on the forget set after the attack, while Acc$_r^*$ is the accuracy on the retained data after the attack.

**Choosing $\alpha$ carefully lets an adversary stay stealthy.** For all baseline MU methods, a moderate range of $\alpha \approx 0.4 - 0.6$ already achieves double-digit or even near-perfect Proto-Acc$_f$ while keeping Acc$_r^*$. Hence an adversary can obtain high forget-class accuracy without noticeably harming the retained classes, making the attack hard to detect.

**Aggressive choice of $\alpha$ exposes the attack.** Setting $\alpha$ too high pushes the patched logits to dominate the original head.

An adversary can recover the pre-unlearning forget accuracy, the noticeable drop in retained accuracy risks betraying the presence of the attack.

**Spotter nullifies the trade-off.** Across the entire sweep, Spotter keeps Proto-Acc$_f$ under $0.3\%$ while preserving Acc$_r^*$ at $\approx 99.96\%$, regardless of $\alpha$. This shows that Spotter's dispersion loss $\mathcal{L}_{\text{sim}}$ effectively breaks the prototype-based attack, eliminating the relearning threat without sacrificing utility.

## D. Baseline Descriptions

*Random Label* (Golatkar et al., 2020) randomly selects labels from retained-classes $c \notin \mathcal{C}_f$ and fine-tune with $\mathcal{D}_f$.

*NegGrad* (Thudi et al., 2022) fine-tunes on $\mathcal{D}_f$, but in the opposite direction, aiming to find the global maximum.

*Boundary Shrink* (Chen et al., 2023) actually removes the targeted decision boundary by selecting the nearest labels adversarially for each data point in $\mathcal{D}_f$ and fine-tunes with label-flipped $\mathcal{D}_f$.

*Boundary Expand* (Chen et al., 2023) adds a temporary "shadow" class—an extra output neuron trained on the forgetting samples—to reroute their activations before pruning it to preserve the original class boundaries.

*SalUn* (Fan et al., 2024) constructs a saliency weight map by identifying gradients that exceed a threshold $\gamma$ using the input data $\mathcal{D}_f$. This saliency weight map determines which model parameters should be updated or preserved when applying MU approaches such as Random Label, NegGrad, and others.

*Learn to Unlearn* (Cha et al., 2024) fine-tunes the model to misclassify the forget set, using adversarial augmentation and the weight penalty to leave other knowledge intact.

*DELETE* (Zhou et al., 2025) applies a mask that adds $-\infty$ to the forgotten-class logits and then softmax normalizes the remaining logits to jointly optimize forgetting and retention without using any retained data.

*Fisher Forgetting* (Golatkar et al., 2020) leverages the Fisher Information Matrix (FIM) to "scrub" model parameters to remove information about the data to be forgotten.

*UNSC* (Chen et al., 2024) primarily focuses on identifying the null space of the class to forget using layer-wise Singular Value Decomposition (SVD). It then projects the subspace corresponding to the retained classes onto this null space, unlearning within that space. During the unlearning process, UNSC calibrates the decision boundary based on the retained dataset to preserve generalization performance.

## E. Spotter on Larger and Real-World Datasets

*Table 8.* Comparisons of the unlearning performances among **larger and real-world datasets.**

| Dataset | Method | Acc$_f$↓ | Acc$_r$↑ | OU @$\varepsilon$↓ | Proto-Acc$_f$↓ | Acc$_r^*$↑ |
|---|---|---|---|---|---|---|
| **Tiny-ImageNet** (Forget 10 Class) | Original Model | 99.68 | 99.6 | - | 99.86 | 99.18 |
| | Retrain Model | 0.00 | 99.61 | - | 30.98 | 98.80 |
| | DELETE | 0.44 | 90.31 | 0.2208 | 25.60 | 89.67 |
| | **Spotter** | 0.56 | 98.67 | 0.0273 | 5.42 | 98.63 |
| **CASIA-WebFace** (Forget 500 Class) | Original Model | 99.00 | 99.16 | - | 99.47 | 99.06 |
| | Retrain Model | 0.00 | 98.93 | - | 78.90 | 98.34 |
| | DELETE | 1.28 | 86.83 | 0.5023 | 25.47 | 86.80 |
| | **Spotter** | 0.74 | 98.89 | 0.0201 | 7.12 | 98.19 |

To examine the practicality of Spotter beyond CIFAR, we additionally evaluate it on (i) a more challenging image-

classification benchmark, Tiny-ImageNet, and (ii) a real-world face recognition dataset, CASIA-WebFace. We follow the same `Spotter` configuration as in the main experiments. For Tiny-ImageNet, we unlearn $|\mathcal{C}_f| = 10$ classes; for CASIA-WebFace, we unlearn $|\mathcal{C}_f| = 500$ identities. Both datasets do not provide a dedicated test split.

Table 8 shows that `Spotter` achieves near-complete forgetting while preserving retained utility: it drives $\text{Acc}_f$ close to zero ($0.56\%$ on Tiny ImageNet; $0.74\%$ on CASIA-WebFace) with high $\text{Acc}_r$ ($98.67\%$ and $98.89\%$, respectively), and maintains small over-unlearning (OU@$\varepsilon$ of 0.0273 and 0.0201, respectively). Moreover, the prototype-based relearning signal remains low (Proto-Acc$_f$ of 5.42 and 7.12, respectively), indicating that `Spotter` continues to suppress class-level residual traces even at a larger scale and in identity-centric data. Overall, these results support the scalability and practical effectiveness of `Spotter` on larger and real-world datasets.

## F. `Spotter` on Transformer Classifiers

*Table 9.* Comparisons of the unlearning performances among various methods with **transformer-based vision models.**

| Model | Method | CIFAR-10 | | | | | | |
|-------|--------|--------|--------|--------|--------|--------|--------|--------|
| | | $\text{Acc}_f\downarrow$ | $\text{Acc}_r\uparrow$ | $\text{Acc}_{ft}\downarrow$ | $\text{Acc}_{rt}\uparrow$ | OU@$\varepsilon\downarrow$ | Proto-Acc$_f\downarrow$ | $\text{Acc}_r^*\uparrow$ |
| **ViT-S** | Original Model | 86.62 | 85.10 | 75.30 | 71.03 | - | 91.48 | 84.12 |
| | Retrain Model | 0.00 | 87.99 | 0.00 | 75.48 | - | 14.00 | 87.56 |
| | **`Spotter`** | 0.30 | 85.19 | 0.40 | 71.69 | 0.0029 | 5.52 | 85.37 |
| **DeiT-S** | Original Model | 87.92 | 84.78 | 76.10 | 71.12 | - | 92.94 | 84.20 |
| | Retrain Model | 0.00 | 87.15 | 0.00 | 73.59 | - | 27.12 | 86.52 |
| | **`Spotter`** | 0.50 | 85.35 | 0.10 | 72.46 | 0.0047 | 3.20 | 85.26 |

To verify that our findings are not tied to a single backbone, we replicated the CIFAR-10 experiments on two modern transformer classifiers—ViT-Small (ViT-S) (Dosovitskiy et al., 2021) and DeiT-Small (DeiT-S) (Touvron et al., 2021)—and summarize the results in Table 9. We extend the Prototypical Relearning Attack to a ViT architecture with minimal changes: we replace the globally averaged convolutional features with the ViT [CLS] token embedding to form class-level prototypes. `Spotter` maintains the utility of transformer-based classifier models with no observable degradation, while driving down performance on the class to be forgotten. OU@$\varepsilon$ score remains close to zero, indicating that it successfully mitigates over-unlearning even in the patch-token space of transformers. In addition, the Prototypical Relearning Attack fails, unable to recover meaningful accuracy on the forgotten classes.

## G. Relearning Attack on SalUn + `Spotter`

*Table 10.* Attack Performance comparison across attack scenarios.

| Method | Proto1 | | Proto5 | | Proto10 | | Proto100 | |
|--------|--------|--------|--------|--------|--------|--------|--------|--------|
| | Proto-Acc$_f\downarrow$ | $\text{Acc}_r^*\uparrow$ | Proto-Acc$_f\downarrow$ | $\text{Acc}_r^*\uparrow$ | Proto-Acc$_f\downarrow$ | $\text{Acc}_r^*\uparrow$ | Proto-Acc$_f\downarrow$ | $\text{Acc}_r^*\uparrow$ |
| SalUn + **`Spotter`** | 0.76 | 98.38 | 1.32 | 98.3 | 1.38 | 98.3 | 2.26 | 98.13 |

To verify our explanation we reran the attack 500 times for each attack scenario (Proto-1, 5, 10, 100) on the SalUn + `Spotter` unlearned model and report the highest Proto-Acc$_f$ observed in the Table 10. The results show that despite being exposed to 500 times attack, SalUn + `Spotter` still demonstrated strong robustness against the Prototypical Relearning Attack. Even when using up to 100 attack samples, SalUn + `Spotter` remains robust to PRA, with Proto-Acc$_f$ staying around 2%.

## H. Sequential Unlearning Scenarios

*Table 11.* **Sequential unlearning with `Spotter` on CIFAR-100** (10 rounds, 1 class per round).

| Round | $Acc_f\downarrow$ | $Acc_r\uparrow$ | $OU@\varepsilon\downarrow$ | $Proto\text{-}Acc_f\downarrow$ | $Acc_r^*\uparrow$ |
|:---:|:---:|:---:|:---:|:---:|:---:|
| 1 | 0.00 | 99.50 | 0.0448 | 2.40 | 99.48 |
| 2 | 0.30 | 99.95 | 0.0239 | 4.10 | 99.94 |
| 3 | 0.20 | 99.97 | 0.0168 | 1.87 | 99.96 |
| 4 | 0.05 | 99.96 | 0.0195 | 4.35 | 99.96 |
| 5 | 0.08 | 99.97 | 0.0163 | 4.64 | 99.96 |
| 6 | 0.00 | 99.98 | 0.0139 | 3.70 | 99.97 |
| 7 | 0.00 | 99.97 | 0.0162 | 3.26 | 99.97 |
| 8 | 0.03 | 99.97 | 0.0151 | 3.40 | 99.97 |
| 9 | 0.07 | 99.98 | 0.0141 | 2.40 | 99.97 |
| 10 | 0.03 | 99.98 | 0.0139 | 2.92 | 99.98 |

Sequential unlearning closely matches real-world deployments, where a service may receive repeated deletion requests over time (often for different users/identities/classes), and each new request must be honored without eroding previously retained capabilities. We further evaluate `Spotter` under a sequential unlearning setting on CIFAR-100, performing 10 rounds of unlearning with one forget class per round. Table 11 shows that `Spotter` prevents catastrophic forgetting across rounds: $Acc_f$ stays near zero throughout, while $Acc_r$ and $Acc_r^*$ remain consistently high and stable. At the same time, $OU@\varepsilon$ remains small and generally decreases over rounds, and $Proto\text{-}Acc_f$ stays low, indicating limited class-level residual traces even after repeated unlearning requests. Together, these results highlight `Spotter` as an effective solution for long-horizon, continuously operating systems that must support unlearning as an ongoing service.

## I. Effect of Forget-Class Ratio

*Table 12.* **Severe class-level unlearning on CIFAR-100.** Performance of `Spotter` when unlearning an increasing fraction of classes (10% / 30% / 50%).

| Setting | $Acc_f\downarrow$ | $Acc_r\uparrow$ | $OU@\varepsilon\downarrow$ | $Proto\text{-}Acc_f\downarrow$ | $Acc_r^*\uparrow$ |
|:---:|:---:|:---:|:---:|:---:|:---:|
| Spotter (10%, main) | 0.00 | 99.96 | 0.0314 | 3.00 | 99.69 |
| Spotter (30%) | 0.00 | 99.95 | 0.0195 | 2.77 | 99.85 |
| Spotter (50%) | 0.00 | 99.94 | 0.0118 | 1.67 | 97.29 |

We further stress-test `Spotter` on CIFAR-100 by unlearning a large fraction of classes (30% and 50%), beyond the default 10% setting in the main paper. This scenario is practically relevant for long-lived services that may receive a large volume of deletion requests, where the unlearning workload can become increasingly severe. Table 12 shows that `Spotter` maintains complete forgetting across all settings while keeping high retained accuracy. Notably, $OU@\varepsilon$ decreases as the unlearning fraction increases, and $Proto\text{-}Acc_f$ remains low, indicating that `Spotter` continues to suppress class-level residual traces even under aggressive unlearning. In the 50% unlearning setting, `Spotter` still preserves strong retained performance overall, supporting its effectiveness in severe unlearning regimes.

## J. Experiments Configuration

### J.1. Hardware Configuration

We conducted our experiments on a server with an NVIDIA A5000 GPU, Intel Xeon Gold processors, and 256 GB RAM.

## J.2. Software Environment

- Operating System: Ubuntu 22.04.3 LTS

- Deep Learning Framework: PyTorch 2.1.1

- Other Dependencies: CUDA 12.1, Python 3.10

