# OpenReview forum: "Unlearning’s Blind Spots: Over‑Unlearning and Prototypical Relearning Attack"
_ICML.cc/2026/Conference — ICML 2026 regular_

### Official Review · Reviewer_RoNx · 2026-03-10

**Soundness:** 3
**Presentation:** 3
**Significance:** 3
**Originality:** 3
**Overall Recommendation:** 4
**Confidence:** 3

**Summary:**

This paper studies machine unlearning problem. Existing algorithms suffer from two issues: (1) forgetting harms nearby retained data (2)relearning attacks where forgotten knowledge can be quickly recovered. The proposed Spotter method can mitigate both issues and improve robustness across several image classification benchmarks.

Experimental results on CIFAR-10/100 show the effectiveness of the proposed method.

**Compliance With Llm Reviewing Policy:**

Affirmed.

**Final Justification:**

The responses address most of my concerns. Thus, I'm learning to accept the paper.

**Key Questions For Authors:**

See above weaknesses.

**Limitations:**

yes

**Strengths And Weaknesses:**

Strengths:
(1) The proposed relearning attack is simple but effective.
(2) The method is flexible and can be integrated into existing unlearning methods.

Weaknesses:
(1) Limited theoretical guarantees. Although the paper empirically confirms the effectiveness of the proposed Prototypical Relearning Attack (PRA) and Spotter algorithm, there is no theoretical analysis showing that the Spotter can guarantee the removal of information or prevent relearning beyond the PRA attack.

(2) The experiments are only conducted on relatively small datasets such as CIFAR-10 and CIFAR-100, with limited validation on larger or more realistic benchmarks. It remains unclear whether the proposed method would scale effectively to large-scale datasets or foundation models.

It would be better to show the effectiveness of the proposed method on large-scale data, like ImageNet, and foundation models, like CLIP.

(3) Does the unlearned model keep the classifier weights for the forget classes?
    How to get $w_{c}^{(u)}$ in Eq.(7)?

(4) It would be better to analyze the connections and differences between the proposed relearning attack and existing relearning attacks.
    And explain why the proposed PRA attack is more powerful.

(5) For the ablations, it would be better to show the conclusion is consistent across different datasets.

---

> ### Author Rebuttal · Authors · 2026-03-30
>
> We sincerely appreciate the reviewer’s positive assessment and helpful suggestions. Encouraged by feedback, we now address the specific concerns below.
>
> **W1. Limited Theoretical Guarantees.**
> We agree that our current work does not provide a formal guarantee that Spotter completely removes all target information or prevents every possible relearning strategy. Our intent is more practical: rather than proposing a certified unlearning framework, we study two concrete blind spots that arise in existing practical unlearning methods—over-unlearning and post-hoc relearning—and introduce both an attack (PRA) and a mitigation objective (Spotter) for them. We will clarify this scope more explicitly in the revision. As future work, we believe it would be valuable to connect these empirical failure modes to certified unlearning frameworks and investigate stronger formal guarantees against broader relearning families.
>
> **W2&W5. Validation on Different Datasets/Models.**
> To directly address the reviewer’s request for validation on ImageNet and a foundation-style model, we evaluated Spotter on a pretrained CLIP ViT-B/32. Specifically, we kept the text encoder fixed and updated only the visual encoder during unlearning, so that the class predictions were still produced in the usual CLIP manner via the inner product between image embeddings and text-prompt embeddings. For this CLIP experiment, we computed OU@ε using the same definition as in the main paper, with the perturbed samples $x_p$ generated by PGD that maximizes the cross-entropy loss. We randomly selected 10 ImageNet classes as the forget set and performed class unlearning with Spotter in this setting. The resulting performance is summarized below:
>
> | Method | $Acc_{ft}$ ↓ | $Acc_{rt}$ ↑ | $OU@\epsilon$ ↓ |
> | --- | --- | --- | --- |
> | Original | 54.18% | 55.75% | -- |
> | Spotter | 0.00% | 55.00% | 0.0141 |
>
> These results strengthen our claim that Spotter is not limited to CNN classifiers. Even in a pretrained vision-language model on a large-scale dataset, Spotter almost perfectly forgets the target classes while preserving retained performance and still maintains a low OU@ε. To strengthen the paper, we will include in the revision an additional experiment on pretrained CLIP in ImageNet classification setting.
>
> Moreover, in Appendix D and E, we evaluate Spotter on various datasets(Tiny-ImageNet and CASIA-WebFace) and transformer-based models(ViT-S, DeiT-S). In both cases, Spotter achieves near-complete forget accuracy and maintains strong retained performance while substantially reducing both OU@ε and vulnerability on PRA.
>
> **W3. How to get $w^{(u)}_{c}$ in Eq. (7)?**
> The unlearned model keeps the class-head structure, including the rows corresponding to the forget classes. PRA constructs prototype-based logits for the forget classes and interpolates them with the corresponding rows of the unlearned classifier, as described in Eq. (7) and Algorithm 1. In other words, PRA is applied to the already-unlearned model $f\_{\theta\_{u}}$, and  is directly read from that model’s final classifier parameters; it is not the pre-unlearning/original weight.
>
> If one instead discards the forget-class rows entirely, PRA can still be instantiated by using $\alpha=1$, i.e., relying only on the prototype-based logits without $w^{u}\_c$. Our Table 5 shows that this $\alpha=1$ case is often the strongest attack on the forget class for vulnerable baselines, but it also tends to incur a noticeable drop in retained accuracy. Therefore, we keep Eq. (7) in its interpolated form for two reasons. First, it matches a realistic white-box audit setting in which the attacker (e.g., the service operator) can inspect the deployed unlearned model parameters and read the forget-class rows from the unlearned head. Hu et al [1] has already shown that post-unlearning recovery can occur even with access to only a small and loosely related relearning set. Second, interpolation is important because PRA is meant to study **a strong yet stealthy** post-unlearning attack.
>
> [1] Hu et al. Unlearning or obfuscating? jogging the memory of unlearned llms via benign relearning. ICLR 2025.
>
> **W4. Why the PRA is More Powerful?**
> To the best of our knowledge, prior work has treated relearning as recovering unlearned knowledge through a small amount of post-hoc fine-tuning (especially in LLM unlearning). PRA, in contrast, is designed specifically for the class-level unlearning setting and exploits a different vulnerability, namely the residual prototype structure left in the unlearned feature space. In other words, PRA is the first non-fine-tuning relearning attack tailored to class-level unlearning.
>
> Figure 2 provides this comparison explicitly. Figure 2-(a) shows that standard fine-tuning-based relearning does not effectively recover forget-class performance in this setting, whereas Figure 2-(b) shows that PRA recovers it much more strongly with only a few samples, while keeping the retained-accuracy drop below 1%.

---

> > ### Author Rebuttal · Reviewer_RoNx · 2026-04-03
> >
> > Thanks for the responses from the authors. It addresses all of my concerns.
> > I'm learning to accept the paper and keep my score.

---

> > > ### Author Response · Authors · 2026-04-03
> > >
> > > We sincerely thank the reviewer for the positive evaluation and encouraging support of our paper. We are very pleased that our rebuttal was able to fully resolve your concerns, and we deeply appreciate your thoughtful feedback.

---

### Official Review · Reviewer_JaZM · 2026-03-13

**Soundness:** 2
**Presentation:** 2
**Significance:** 2
**Originality:** 2
**Overall Recommendation:** 3
**Confidence:** 5

**Summary:**

This paper focuses on two “blind spots” of machine unlearning in class-level forgetting scenarios: (1) over-unlearning, that is, the forgetting process causes collateral damage to the retained samples near the boundary of the forgotten class; (2) relearning attack, that is, the attacker uses a very small number of forgotten class samples to quickly recover the deleted knowledge.
The authors propose two core technical points: First, they define a new evaluation metric, OU@ε, attempting to quantify "excessive forgetting" in the neighborhood of forgotten classes without accessing retained data; second, they propose a novel attack, Prototypical Relearning Attack (PRA), which utilizes the prototype of the forgotten class residual feature clusters to recover classification ability. To address both problems, the authors further propose Spotter, which combines masked distillation on the boundary neighborhood with the loss of forgotten class feature dispersion to simultaneously reduce OU@ε and improve robustness to PRA. Experiments cover CIFAR-10/100, Tiny-ImageNet, CASIA-WebFace, and Transformer backbone, and the results show that Spotter outperforms several existing methods under the authors' defined metrics.

**Compliance With Llm Reviewing Policy:**

Affirmed.

**Final Justification:**

I will keep my score considering the limited contribution and the application of the proposed method (Q6).

**Key Questions For Authors:**

See weakness

**Limitations:**

The author did not discuss limitations of their paper.

**Strengths And Weaknesses:**

Strength:
1. This article goes beyond the crude metric of decreased forgetting accuracy, pointing out two deeper failure modes of machine forgetting: local collateral damage and the risk of post-event recovery. This is valuable and closer to real-world deployment needs than many works that merely pursue forgetting accuracy close to 0.
2. Plug-and-play mitigation design. Spotter is designed as an add-on objective and is shown combined with several baselines, which increases its practical relevance.
3. The paper includes multiple datasets, including a face-recognition setting, and extends results to ViT models.
4. The writing is good.

Weakness:
1. Two “blind spots” of machine unlearning are not first proposed by the paper. Many existing works focus on these points, which limit the contribution of this paper.
2. Spotter appears to be a combination of known ingredients: local distillation and feature dispersion. The novelty lies more in the application framing than in a fundamentally new algorithmic idea. For a top-tier venue, the paper should better clarify why this combination is non-trivial and why existing regularization ideas do not already imply it.
3. The semantic validity of OU@ε is not sufficiently established. OU@ε measures divergence between the original model’s masked distribution and the unlearned model’s output on perturbations around forget samples. However, it is not clear that this quantity cleanly corresponds to over-unlearning. Points near the forget set may still semantically belong to the forget concept, so changes there may reflect desired forgetting rather than collateral damage.
The metric assumes that the original model’s non-forget relative probabilities in this region are the correct target to preserve, which is a strong and under-justified assumption.
As a result, OU@ε seems closer to a local output drift metric than a validated measure of over-unlearning.

4. If the paper aims to propose a new evaluation metric, it should provide stronger evidence that OU@ε correlates with actual degradation on retained examples near the forget boundary. Right now, the paper mainly shows that Spotter reduces OU@ε, but not that OU@ε is a trustworthy proxy for the phenomenon it claims to capture.
5. Comparisons may favor the proposed method by construction.
Spotter explicitly optimizes against the paper’s own metric (OU@ε) and own attack (PRA), while baselines were not designed for these objectives. This does not invalidate the results, but it limits the strength of the empirical claims. I would like to see stronger evaluation under independent criteria and alternative attacks.
6. The threat model for PRA is underspecified and somewhat inconsistent. The paper alternates between an attacker who is the service operator with white-box access and a practical scenario where a few public images are available. These are very different capabilities.
If the attacker truly has white-box access and can patch the classifier head, stronger attacks than PRA may already be available.
If the attacker is external, it is unclear whether they can obtain embeddings or modify logits as assumed.
The paper would benefit from a clearer separation of white-box audit attack vs realistic external attack.

---

> ### Author Rebuttal · Authors · 2026-03-30
>
> Thanks for reviewer's insightful comments. We address the specific concerns below.
>
> **W1.** We agree that the general concerns of over-unlearning and relearning risk are not introduced for the first time by our paper. Rather, our intended claim is more specific.
> First, to the best of our knowledge, **this work is the first to explicitly formalize and quantify over-unlearning in the class-level unlearning setting through a dedicated metric, OU@ε.** Prior work has discussed utility degradation or side effects, but has not directly operationalized over-unlearning as a measurable target in this setting. This distinction is important because our contribution is not merely to point out the phenomenon, but to make it evaluable, and optimizable.
> Second, while relearning has been studied mainly in LLM unlearning through few-shot fine-tuning-based recovery with white-box access, it has received much less attention in class-level unlearning. In this context, PRA is, to the best of our knowledge, the first relearning attack that explicitly exploits residual feature-distribution structure. Thus, our contribution is not to introduce these concerns, but to make them explicit and actionable.
>
> **W2.** Our contribution is not that local distillation or dispersion are individually new, but that their combination is driven by two newly operationalized failure modes that prior unlearning methods do not explicitly address. Local distillation is used not as a generic retention regularizer, but to control boundary-proximal over-unlearning captured by OU@ε. Likewise, PRA shows that even after class-level unlearning, the residual feature structure of the forget class can remain exploitable; this motivates our dispersion term, which aims to disrupt that residual geometry rather than merely reduce forget accuracy. Thus, the novelty is not in introducing new primitive components, but in investigating these blind spots, turning them into concrete optimization targets, and designing a joint objective to address them.
>
> **W3&W4.** OU@ε is intended to measure collateral distortion of retained knowledge in the forget-adjacent region, not arbitrary output change around forget samples. Concretely, for a perturbed sample $x_p$ around the forget set, it masks out the forget class and evaluates whether the relative relationships among the non-forget classes are preserved after unlearning. Thus, even when $x_p$ remains semantically close to the forget class, OU@ε does not ask the model to preserve forget-class evidence; it asks whether unlearning unnecessarily perturbs the local structure of the retained classes. This is why we define OU@ε on the masked retained-class distribution and interpret it as a boundary-proximal measure of over-unlearning.
>
> To validate that OU@ε reflects actual collateral damage, we additionally evaluated retained test examples closest to the forget prototype in representation space and measured Top-n% retain-subset accuracy. Across methods, OU@ε shows a strong negative correlation with this local retained accuracy (Pearson correlation coefficients $r ≈ -0.81 / -0.84 / -0.82$ for Top-2/5/10%), notably stronger than with overall retained accuracy $Acc_{rt}$ ($r ≈ -0.70$). This matters because prior class-level unlearning evaluations often use global retained accuracy as a naive proxy for over-unlearning, whereas our results show that OU@ε better tracks degradation on retained examples most entangled with the forget class.
>
> Link: https://imgland.net/i/ExVzXBLO/_.png
>
> **W5.** The above results support OU@ε captures meaningful local collateral damage rather than abstract output drift. Accordingly, Spotter is not merely optimized for a self-defined metric; we introduce the OU-aware objective because OU@ε empirically aligns with local retained-example degradation, and Spotter reduces that degradation. For PRA, our intent is not to evaluate Spotter on a convenient self-defined attack. Rather, PRA is introduced because, in this class-level setting, few-shot fine-tuning can be too weak and may create a false sense of security. PRA exposes a novel vulnerability, and Fig. 2 shows that it recovers forget-class performance much more effectively than prior relearning. Spotter is designed to mitigate this stronger risk with the loss that spread embeddings to make them robust against the threats.
>
> **W6.** We acknowledge that our threat model needs to be clarified. PRA is designed as a white-box audit attack using only a small subset of forget-class samples, consistent with prior fine-tuning-based relearning works [1]. While stronger white-box attacks may exist, this is precisely our point: even a minimal, low-cost attack using only a handful of samples is sufficient to significantly recover performance on the forgotten class, highlighting a fundamental vulnerability in existing class-level unlearning methods.
>
> [1] Hu et al. Unlearning or obfuscating? jogging the memory of unlearned llms via benign relearning. ICLR 2025.

---

> > ### Author Rebuttal · Reviewer_JaZM · 2026-04-04
> >
> > Thanks for your response.
> > 1. There are some existing unlearning work uses PRA.
> > 2. The contribution is limited. The technical contribution is limited, and the contribution of investigation is not sufficient.
> > 3. Q6 is not well addressed, limiting the contribution and setting of this work.
> > 4. The rebuttal of Q5 is not enough; I'd like to see some experimental results.

---

> > > ### Author Response · Authors · 2026-04-06
> > >
> > > We thank the reviewer again for the follow-up comments. We would like to offer three brief clarifications.
> > >
> > > First, regarding PRA, we would like to state this point more explicitly. We are not aware of any prior machine-unlearning work that proposes the specific attack we call Prototypical Relearning Attack (PRA). Prior relearning studies have mainly focused on fine-tuning-based recovery procedures, whereas PRA is a distinct prototype-based recovery attack that directly exploits residual feature structure by constructing forget-class prototypes from only a few forget samples and using them to restore forgotten-class decision ability. In that precise sense, PRA is our proposed attack as the reviewer mentioned in Summary, not a reused attack protocol from prior work. For this reason, without a concrete reference, it was difficult for us to understand precisely which prior work the reviewer had in mind.
> > >
> > > Second, on Q5, our rebuttal aimed to provide an evaluation beyond our proposed metric alone. In particular, we added results on retained test samples closest to the forget prototype in representation space, and Spotter showed the strongest behavior on these highly entangled local subsets (Top-2/5/10%). Here's the results that we presented in rebuttal: [Link - https://imgland.net/i/ExVzXBLO/_.png]. OU@ε revealed a stronger correlation with the Top n% score compared to the retain performance. We believe this supports that the benefit is not limited to optimizing OU@ε by construction, but also appears under an independent near-forget retention criterion. In addition, Fig. 2-(a) already shows that Spotter remains more robust than prior methods under conventional fine-tuning-based relearning, which was precisely why we introduced PRA as a stronger and more targeted audit attack.
> > >
> > > Third, on Q6, our intention was to clarify PRA as a white-box audit attack with a very small number of forget-class samples. The goal is not to claim that PRA is the strongest possible white-box attack, but to show that even a simple, low-cost recovery procedure can substantially revive forgotten-class performance after unlearning. In that sense, we view PRA as a practically meaningful vulnerability test for class-level unlearning systems.
> > >
> > > We believe there may have been some mismatch between our intended scope and how our contribution was interpreted. In particular, we hope the reviewer will recognize that our contribution lies in explicitly and simultaneously addressing two themes that have become important in recent unlearning research: formalizing and quantifying over-unlearning in the class-level setting as an optimization target, and introducing a new relearning threat that goes beyond the conventional paradigm.

---

### Official Review · Reviewer_2tqb · 2026-03-13

**Soundness:** 2
**Presentation:** 3
**Significance:** 2
**Originality:** 2
**Overall Recommendation:** 4
**Confidence:** 4

**Summary:**

The authors have identified two key issues in the existing stream of works in machine unlearning, namely: over-unlearning and vulnerability to distribution based relearning attacks. Furthermore, they claim that there is no valid metric to quantify over-unlearning during machine-unlearning. So firstly, the authors have suggested a metric named $OU@\epsilon$, which is based on computing the divergence between original model’s masked output probabilities and the unlearned model’s probabilities on perturbed samples of the forget set. Secondly, they also suggest a relearning attack that exploits the  clustering of output representations of the forget set. Finally, they suggest a “plug-and-play” method called SPOTTER to counter over-unlearning and distribution based relearning attacks. SPOTTER is a combination of three objectives, which: 1. distill knowledge from original model to learn on regular classes and reducing knowledge from forget classes, 2. Counter over-unlearning by applying the same loss on a perturbed forget set, and 3. disperse the embeddings of the forget class by computing the averaged pairwise cosine similarities between the samples of the unlearned feature embeddings.

**Compliance With Llm Reviewing Policy:**

Affirmed.

**Key Questions For Authors:**

1- Looking at the results, while using SPOTTER with other methods does seem to increase the performance of those methods, however, the performance of the methods in combination with SPOTTER is still worse than using just SPOTTER. Considering this, would it still be okay to present it as a “plug-and-play” method.



2- In section, 3.1. after equation (3), the authors have stated “D(·∥·) is a divergence measure between two probability vectors, such as Kullback-Leibler (KL) and JensenShannon (JS) divergences.” Later, it has been mentioned that JS divergence has been used for the $OU@\epsilon$ metric. How was the descision made? What differences could be found in using either of those two?

**Limitations:**

yes

**Strengths And Weaknesses:**

Strengths:


The suggestion of the metric $OU@\epsilon$ to properly quantify over-unlearning is both novel and well-motivated, particularly given that prior works on Machine Unlearning have not addressed this.

Using the representations of samples of the forget set to perform a relearning attack is very clever and intuitive.

The paper presents a extensive set of experiments to prove the claims that have been made.


Weakness:


There are no experiments to visualize the representation after adding the perturbations to the instances of forget set. Adding this would help with visualizing the pertubations is actually around the
The authors define SPOTTER to be a “plug-and-play” objective. But later define only the first objective of SPOTTER is “plug-and-play”. This is very misleading.


In Table 1, the $OU@\epsilon$ score of Boundary Expand method on CIFAR-100 is shown to be the least (0.0043) . Based on the description of the results, it seems to be a typo.

---

> ### Author Rebuttal · Authors · 2026-03-30
>
> We thank the reviewer for the positive assessment and for recognizing the novelty of OU@ε and the intuition of the relearning attack. We also appreciate the constructive suggestions. Detailed responses to each point are provided below.
>
> **W1. Visualization of Perturbations around the Forget Set.**
> We thank the reviewer for this helpful suggestion. We agree that a direct visualization of the perturbed forget samples can make our notion of boundary-proximal over-unlearning more intuitive. In the revision, we will add a representation-space visualization comparing clean forget samples with their perturbed counterparts generated by PGD and Gaussian noise. Consistent with our formulation, $A\_\varepsilon(D\_f)$ is intended to capture local perturbations around forget examples, and our primary instantiation uses PGD specifically because it more strongly pushes samples toward the decision boundary, whereas Gaussian noise serves as a local but less targeted control. As illustrated by the attached PCA plots, the PGD-perturbed forget samples are displaced in a more directional and expanded manner relative to the clean forget samples, while Gaussian perturbations remain comparatively more isotropic and locally scattered. We do not claim that a 2D PCA plot can perfectly certify the exact boundary location, but this visualization provides qualitative evidence that PGD perturbations move forget samples toward the boundary-relevant region more effectively than untargeted random noise.
>
> Link: https://imgland.net/i/9pJXq8eD/_.png
>
> **W2&Q1. Claim on the "plug-and-play" Compatibility.**
> The reviewer raises an important point: our current wording overstates the scope of the plug-and-play claim and can be read as if the entire Spotter objective were universally plug-and-play. The intended claim is narrower and should consistently follow the formulation already stated in Sec. 4: “our Spotter’s first term $\mathcal{L}\_u$ offers plug-and-play compatibility by allowing the base unlearning loss to be replaced with an existing method, while $\mathcal{L}\_o$ and $\mathcal{L}\_{\mathrm{sim}}$ are then added to mitigate over-unlearning and PRA, respectively.
> We will also revise the abstract, introduction, and conclusion accordingly (which we used the term "plug-and-play objective").
>
> Under this more precise interpretation, the value of the compatibility claim is not that every baseline + Spotter combination must outperform standalone Spotter on all metrics. Rather, it is that the proposed regularizers can be modularly attached to an existing unlearning pipeline to address two blind spots--over-unlearning and PRA vulnerability--without requiring retained data or architecture-specific redesign. This is the role of Table 2: when applied to representative baselines, Spotter consistently and substantially reduces both OU@ε and Proto-Acc$\_f$ by large margins.
>
> This also explains why standalone Spotter can remain stronger than some Spotter-augmented baselines. Standalone Spotter evaluates the strength of our full design, including our own base loss $\mathcal{L}\_u,$ whereas the combined variants test whether the proposed add-on terms can strengthen existing methods with minimal modification. In this sense, stronger standalone performance does not contradict the compatibility claim; rather, it indicates that our default $\mathcal{L}\_u$ is itself a strong base objective, while the combined variants still inherit some limitations of the original baseline.
>
> More broadly, this compatibility is valuable beyond the current baselines. Stronger future unlearning objectives may also be integrated into the Spotter's first term. In that case, $\mathcal{L}\_o$ and $\mathcal{L}\_{sim}$ can continue to play the role of targeted regularizers for over-unlearning and PRA.
>
> **W3. OU@ε of Boundary Expand.**
> We thank the reviewer for the careful reading and for pointing out the typo. We will revise the description of the Boundary Expand results in the Sec 5.2.
>
> **Q2. Selection of Divergence Measure.**
> Our formulation of OU@ε is divergence-agnostic; Eq. (3) intentionally allows either KL or JS. We use KL in unlearning because it is a standard distillation objective and provides a simple optimization target for unlearning. For evaluation, we use JS because it is symmetric and $[0,1]$ bounded, which makes the scale of OU@ε more stable and interpretable across methods and datasets. We will clarify this unlearning vs. evaluation distinction in the revision.

---

### Official Review · Reviewer_aCe7 · 2026-03-13

**Soundness:** 2
**Presentation:** 3
**Significance:** 3
**Originality:** 4
**Overall Recommendation:** 4
**Confidence:** 4

**Summary:**

This paper investigate over-unlearning and relearning in the context of classification model. The paper first propose a metric for over-unlearning and a new type of relearning attack. Based on these, the paper then propose a new unlearning method called Spotter to suppress both over unlearning and prototypical relearning. Empirical results successfully alleviates these two threats.

**Compliance With Llm Reviewing Policy:**

Affirmed.

**Final Justification:**

The author addressed my concerns and promised to provide clarity and explanations to sections that introduce confusions

**Key Questions For Authors:**

See weakness

**Limitations:**

yes

**Strengths And Weaknesses:**

Strengths:
- The paper is well motivated and understanding over unlearning and relearning are very important study in this community.
- The ideas are novel, especially the Prototypical Relearning Attack. The attack seems to significantly improves over prior relearning efforts for per-class unlearning.

Weaknesses:
- [Major] A major issue of this paper is the purpose of unlearning. The (standard) goal of unlearning is to remove the effect of training data from learned model. Therefore, the resulting model should behave as if it never sees the forget set (retrained model). The main reason that retrained model is not a gold standard in the GenAI world is mainly because one specific goal in the GenAI world is to prevent generation of unwanted material. So the goal for "unlearning" is more like an alignment rather than removing effect of *training* data. Now in this classification setting studied in this paper, it is unclear what is the goal of unlearning. If the goal is standard unlearning goal, then the proposed method seems to fail on the Proto-Acc test as suggested in Table 1. Imagine a game where an adversary wants to infer which model is retrained and which model is unlearned, the adversary can simply run the Prototypical Relearning Attack and make a guess with high confidence. If the goal is to absolutely make the model fail on one class via a post-processing technique, I think the method stands as a good contribution, but the authors need to provide some motivating examples on why such scenario is practical.
- [Major] Technically, one major issue is the the unlearning setting is a bit restricted. The evaluation method and relearning attack both seem to be dependent on unlearning happening at class-level, which is not directly clear to me why it is practical. A broader scenario of unlearning subsets across classes is not captured by the current methodology.
- [Major] It's unclear to me why Over Unlearning is defined via an adversarial example type of definition. Why does the distance to the boundary matters here. It only seems reasonable if the goal of unlearning is the latter in weakness 1, make the model fail on one class entirely.
- [Minor] The author should also add retrain model's $OU_{@\epsilon}$ value as retaining is essentially exact unlearning.

Overall I think this paper provides some interesting contribution. But at its current form, the major thing is that the unlearning goal is unclear, which could easily confuse the readers on how to put the value of this paper.

---

> ### Author Rebuttal · Authors · 2026-03-30
>
> We sincerely thank the reviewer for the thoughtful and constructive feedback. We address your feedback in detail as follows.
>
> **W1.**
> We thank the reviewer for raising this important point. While retraining without the forget set is a natural reference under standard sample-level unlearning, our paper instead targets privacy- and safety-driven class-/identity-level unlearning, where **the objective is not retrain equivalence but reliable prevention to recognize or recover the forgotten class,** where the object to be removed is not merely a finite set of training samples in forget classes, but the class-level knowledge itself. In this setting, the practically relevant objective is that the model should no longer reliably recognize or recover the forgotten class, including from previously unseen samples of that class.
>
> To show the practicality of our framework, we explicitly discuss the scenarios of face recognition and identity-level unlearning. In such settings, a right-to-be-forgotten is not completely satisfied if the model can still be quickly re-induced to recognize the forgotten identity from only a handful of images. This is exactly why we introduce PRA: in real-world vision deployments, such seed images can be collected from social media or public profiles, making post-unlearning recovery a realistic concern. To make this point clearer, we augment Figure 4 with an additional example of PRA on the retrained model, highlighting that retrain equivalence alone may still be insufficient in privacy-sensitive services.
>
> Link: https://imgland.net/i/g3oVxnFJ/_.png
>
> We agree retraining is still a useful reference for standard forgetting/retention metrics. However, our paper studies an additional objective beyond this standard view: robustness to post-unlearning re-induction. Therefore, we argue that Proto-$Acc_{f}$ cannot be interpreted as a retrain-equivalence metric; rather, it measures a distinct post-unlearning vulnerability. This perspective is also aligned with recent unlearning work in LLMs, e.g., [1], where robustness to relearning is treated as post-unlearning objectives.
>
> [1] Fan et al. Towards LLM Unlearning Resilient to Relearning Attacks: A Sharpness-Aware Minimization Perspective and Beyond. ICML 2025.
>
> **W2.**
> Our work focuses on class-level unlearning, rather than addressing the most general setting of arbitrary subsets spread across classes. This restriction is intentional: the paper focuses on a practically important regime in which the removal target is a class or identity, such as identity deletion in face recognition or suppression of a sensitive visual category. In these settings, the relevant failure modes are also class-level.
>
> To support this argument, we respectfully emphasize Spotter achieves state-of-the-art performance on class-level unlearning addressing two failure modes that prior methods do not explicitly handle, while also performing well under privacy-sensitive identity removal scenarios.
>
> **W3.**
> We thank the reviewer for this important question. We agree that the motivation of OU@ε should be stated more clearly. Our paper does not treat over-unlearning as deviation from a retrained model. Rather, consistent with the practical class-level unlearning goal discussed in our response to W1, our objective is to remove the target class while preserving as much of the original model's knowledge on the retained region as possible.
>
> This perspective is also consistent with recent analysis of unlearning difficulty. In particular, Zhao et al. [2] show that unlearning becomes harder when the forget and retain sets are more entangled in the original embedding space, and explicitly note that when a forget example and a retain example are close neighbors, aggressively altering the former can inadvertently distort the latter as well. This directly supports our focus on the neighborhood around the forget region: it is precisely where forgetting and retention are most tightly coupled, and thus where unnecessary collateral damage is most likely to occur.
>
> This is why OU@ε uses the original model's masked softmax over retained classes as the reference, instead of a retrained model. By masking out the forget classes, OU@ε asks whether the relative predictive structure over the retained classes is preserved near the forget boundary, i.e., whether the method removes the target class without distorting nearby retained knowledge.
>
> [2] Zhao et al. What makes unlearning hard and what to do about it. NeurIPS 2024.
>
> **W4.**
> We agree that reporting the retrained model’s OU@ε can provide additional context, and we will include it in the revised version for completeness. However, as clarified in our response to W3, we do not view retraining as the normative reference for OU@ε. The reason is that retraining and OU@ε capture two different desiderata. Retraining is a useful reference for deletion fidelity. In contrast, OU@ε is designed to measure preservation of the original model’s retained knowledge.

---

> > ### Author Rebuttal · Reviewer_aCe7 · 2026-04-03
> >
> > Thanks for your response.
> > Re W1. Thanks for the clarification. In that case I strongly recommend the authors to very clearly state 1. the goal of unlearning and the definition of "privacy- and safety-driven class-/identity-level unlearning", 2. clearly discuss its difference with standard approximate unlearning definition, 3. motivate the choice of this specific unlearning goal by <i> your reasoning, <ii> related works that uses similar unlearning definition (there are a lot in GenAI space but ideally the authors would like to gain insights in the classification space). This is crucial as it clears the confusion and makes and setting clear and  helps reducing noise introduced to the unlearning community.
> >
> > Re W2. I'm not sure I fully agree the argument "In these settings, the relevant failure modes are also class-level". In many cases identity is an attribute, not a label for some training data. For example, fro CelebA, the actually classification task is usually different from, inferring who this person is. So unlearning such attribute, rather than label, is equally important in my opinion, which is a limitation that is not covered under the current framework. While this is fine, I encourage this to be stated in the discussion / limitation section more clearly.
> >
> > Re W3. I agree with Reviewer JaZm that the naming of OU might cause confusion as it measures retain performance rather than forget performance. The idea of OU itself can be viewed as OU on forget and OU on retain. In this paper the authors refers to the later but with a broad term OU. I recommend fixing this notation.

---

> > > ### Author Response · Authors · 2026-04-04
> > >
> > > **Re W1.**
> > > We thank the reviewer for understanding the direction and motivation of our unlearning objective. We also fully understand the reviewer’s concern that, while this type of post-unlearning objective is relatively familiar in GenAI unlearning, it may not be immediately intuitive in the classification setting without a more explicit conceptual framing.
> > > In the current submission, due to the 8-page limit, we prioritized presenting the methodology, experiments, and empirical motivation as clearly and concretely as possible, and likely did not devote sufficient space to fully formalizing this distinction and its relation to the standard approximate unlearning view. We appreciate that this may leave room for confusion, especially for readers who naturally interpret unlearning primarily through the lens of retrain equivalence.
> > > In the camera-ready version, where an additional page is available, we will revise the paper to address this concern much more explicitly.
> > > In particular, we will:
> > >
> > > 1. Revise the Introduction to clearly state that our paper studies a privacy- and safety-driven class-/identity-level unlearning setting, where the goal is not strict retrain equivalence, but preventing the model from reliably recognizing or recovering the removed class/identity, while preserving retained knowledge as much as possible.
> > >
> > > 2. Expand the related work and discussion sections to better motivate why this objective is practically relevant in privacy-sensitive classification settings, and to connect it more clearly to prior works that study similar post-unlearning robustness concerns.
> > >
> > > 3. Explicitly contrast this setting with standard approximate unlearning in the discussion section.
> > >
> > > **Re W2.**
> > > We agree that this broader attribute-/identity-level removal setting is important. In datasets such as CelebA, the prediction task may concern facial attributes while identity is not itself the class label, and unlearning a person in such cases would not reduce to class-level removal. This case is beyond the scope of the current framework, and we will state this limitation more explicitly in the discussion section.
> > >
> > > **Re W3.**
> > > We would like to clarify that OU@ε is not intended to be a purely retain-side metric. Although it is evaluated on perturbed samples near the forget-boundary, its formulation is also directly affected by whether the forget-class logits are sufficiently suppressed. In particular, if the forget-class logit remains large, probability mass is still assigned to the forget class, which in turn increases the discrepancy measured by OU@ε even when the sample itself belongs to the retained set. Conversely, OU@ε also increases when the retained-class predictive structure around the forget region is unnecessarily distorted. Therefore, OU@ε reflects both insufficient forget suppression and collateral damage on nearby retained knowledge, measured locally around the forget boundary.
> > > We also agree that the current manuscript does not sufficiently emphasize why this boundary-local view is the right place to assess over-unlearning in class-level unlearning (Reviewer JaZm also suggested improving this point). To make this motivation explicit, we will add a experiment in the revised version that evaluates retained test samples closest to the forget prototype (Top-2/5/10% in representation space). As shown in our additional results, many methods maintain very high overall retain accuracy, but their performance drops substantially on these boundary-neighbor retained samples, indicating that the effect of class-level unlearning is concentrated near the forget boundary rather than uniformly across the retained set. This supports our choice of measuring over-unlearning locally around the boundary. We will revise the text accordingly so that the role of OU@ε is stated more precisely and clearly.
> > >
> > > Link: https://www.imgland.net/i/ExVzXBLO/_.png
> > >
> > > We sincerely thank the reviewer again for the thoughtful and constructive follow-up comments. We made our best effort to carefully address all of the concerns raised, and we are especially grateful for the reviewer’s recognition of the merits of our methodology and experimental analysis. We also truly appreciate the reviewer’s detailed suggestions on clarifying the unlearning objective and better motivating OU@ε. These comments helped us substantially improve the framing, presentation, and scope of the paper, and we believe the revised manuscript is significantly stronger as a result.

---

### Decision · Program_Chairs · 2026-04-30

**Decision:**

Accept (regular)

**Comment:**

The paper analyses two fundamental problems (blind points) in class unlearning: over-unlearning and post-hoc relearning attacks. To address these two problems, the paper proposes a method based on local distillation and feature dispersion, both known techniques that are exploited to mitigate these blind points.

The paper further proposes a new metric targeting over-unlearning, which has been questioned by reviewers. Additional evidence is provided in the rebuttal to show that the proposed metric is relevant (though may not be fully equivalent) to the intended evaluation. Another attack, PRA, is clarified to be a white-box auditing attack with a few forget samples.

Overall, the paper provides contributions in proposing a new metric and a threat model targeting these two blind points. The scope of the paper could be further narrowed to class unlearning in the revised version. The scope of the metric and the attack are also need to be clarified. The rebuttal partially addresses the concerns and most reviewers remain positive towards the paper.